# Active tectonics of the onshore Hengchun Fault using UAS DSM combined with ALOS PS-InSAR time series (Southern Taiwan)

Benoit Deffontaines[1,2], Kuo-Jen Chang[2,3], Johann Champenois[4], Kuan-Chuan Lin[5], Chyi-Tyi Lee[2,6], Rou-Fei Chen[2,7], Jyr-Ching Hu[2,5], Samuel Magalhaes[8]

[1]LAREG Unit (IGN-UDD-IPGP-UPEM), LaSTIG (IGN-UPEM), Université Paris-Est, Marne-la-Vallée, 77454, France
[2]Lab. Int. Assoc. D3E N°536, CNRS-MOST France-Taiwan.
[3]Department of Civil Engineering, National Taipei University of Technology, Taipei, 10654, Taiwan
[4]Laboratoire Tectonique et Mécanique de la lithosphère, IPGP, Paris, 75005, France
[5]Department of Geosciences, National Taiwan University, Taipei, 10617, Taiwan
[6]Department of Applied Geology, Central Taiwan University, Chungli, 32001, Taiwan
[7]Departement of Geology, Chinese Culture University, Taipei, 11114, Taiwan
[8]Alphageomega, 62, Rue du Cardinal Lemoine F-75005 Paris, France

*Correspondence to*: Kuo-Jen Chang (epidote@ntut.edu.tw)

**Abstract.** Characterizing active faults and quantify their activity are major concern in Taiwan, especially following the major Chichi earthquake of September 21[st], 1999. Among the targets that still remain poorly known in terms of active tectonics, are the Hengchun and Kenting Faults (Southern Taiwan). From the geodynamic point of view, they affects the outcropping top of the Manila accretionary prism of the Manila subduction zone that runs from Luzon (N. Philippines) to Taiwan. In order to better settle the location and quantify the activity of the Hengchun Fault, we start from the existing geological maps ones, which we update thanks to the use of two products derived from Unmanned Aircraft System's acquisitions: 1) the very high precision (<50 cm) and resolution (<10 cm) Digital Surface Model (DSM), and the 2) the georeferenced aerial photograph mosaic of the studied area. Moreover, the superimposition of the resulting structural sketch map with new Persistent Scatterers (PS) -InSAR results obtained from PALSAR ALOS-1 images, validated by Global Positioning System (GPS) and leveling data, allow characterizing and quantifying the surface displacements during the monitoring period (2007-2011). We confirm herein the geometry, the characterization and the quantification of the active Hengchun Fault deformation that act as an active left-lateral transpressive fault. As the Hengchun ridge faces one of the last major earthquake of Taiwan (December 26[th] 2006, depth: 44 km, $M_L$=7.0), it is needed to better constrain the Hengchun Peninsula active tectonics in order if possible to prevent major destructions in the near future.

## 1 Introduction

The island of Taiwan is the place of the so rapid NW-SE trending oblique Eurasian versus Philippine Sea Plates convergence (e.g. Ho, 1986, 1988). A Southwest propagation of the deformation in Taiwan is proposed by Suppe (1981; 1984). It is confirmed for instance by the continuous GPS measurements that highlight displacements in the Southern Taiwan of about 60 mm y[-1] (e.g. Yu et al., 1997). This local shortening rate is about the maximum range in the world. We focus herein in the

Southern Taiwan area (so called the Hengchun Peninsula), that is geodynamically interpreted as the Taiwan incipient collision zone (Lallemand et al., 2001, Fig.1) which is the transition of the Manila subduction area toward onshore Taiwan collision. The Hengchun Peninsula is situated at the summit of the outcropping onshore Manila accretionnary prism (Malavieille et al., 2002; Chang et al., 2003).

From the geological point of view, it is a complex area, as its lithology is composed of soft both folded turbidites (Mutan formation) as well as the "Kenting Melange" situated close to the Hengchun Fault zone (Fig.1) and composed of highly deformed shales and large blocks of various lithologies. Evidence of this structural complexity are the five different 1/50.000 scale geological maps (Huang et al., 1997, 2006; Chang et al., 2003 and 2009; Zhang et al., 2016) that have been drawn independently and that are geologically incompatible. Two different major fault zones (Hengchun and Kenting Faults)

NNW-SSE trending parallels the eastern side of the 15 km long Hengchun valley and had been recognized by previous authors as potentially active (e.g. Huang, et al., 1997, 2006; Chang, et al., 2003, 2009). In addition, these two faults affect soft, muddy and alluvial deposits and classical microtectonics and structural studies are impossible as there is no clear preserved fault plane and it is impossible to get any tectonic marker. As the Hengchun Peninsula is the place of sensitive industries, such as the TAIPOWER Nuclear Power Plant N°3, it is a major concern to better decipher the neotectonics and

the active tectonics of the place. Consequently, we address in this paper an indirect approach using Unmanned Aircraft/Aerial System (UAS), Digital Elevaion Model (DEM), structural photo-interpretation and quantitative measurements in order to locate, characterize and quantify the active deformation of the Hengchun Fault.

Below, we first focus on the inputs of high resolution Digital Elevation Model (DEM below) and the orthorectified images of the Hengchun Fault Zone acquired from UAS in order to locate active faults. For a terminological point of view, we used

herein the UAS acronym that encompasses all the aspects of deploying those aircrafts and which is less restrictive than Unmanned Aircraft/Aerial Vehicle (UAV) that corresponds only to the platform itself. Then following the classical photo-interpretation techniques (e.g. Deffontaines et al., 1991), we carefully map and interpret the new morphostructures in term of geological mapping, revealing the active tectonic structures using the so precise resolution and precision UAS Digital Surface Model (DSM). In order to characterize and quantify their tectonic activity, we benefit from Persistent Scatterers-

Interferometry Synthetic Aperture Radar (PS-InSAR hereafter) results derived from ALOS-1 radar images that give us the active interseismic displacement along the Line of Sight (LOS herein) of the radar through the 2007-2011 period. The latter is validated by two E-W trending precise leveling lines across the Hengchun Fault and a few GPS stations (see Academia Sinica GPS, with a reference line in between Taipei and the Penghu Island) that give locally the vertical and the absolute 3D displacement respectively. Finally, the discussion leads us to propose a simple geodynamic and active tectonic model for the

Hengchun Fault that fits with all the available data.

## 2 High resolution Digital Surface Model (DSM) obtained from UAS

An Unmanned Aircraft System (UAS), commonly known as a drone, is an aircraft without human pilot on board that flies autonomously, and is now been used widely for many aspects because of its convenience, high resolution, etc. (Huang and Chang, 2014; Fernandez Galarreta et al., 2015 ; Giordan et al., 2015 ; Tokarczyk, 2015 ; Bühler et al., 2016; Deffontaines et al., 2016A). The UAS used in this study is a modified version of the already-available Skywalker X8 delta-wing aircraft reinforced by carbon fiber rods and covered partially by Kevlar fiber sheets. The drone is launched by hand, flies and takes photos autonomously, then glides back down to the ground by using a pre-programmed flight plans organized by ground control system, and controlled by the ground control station and remote controller. The autopilot system used in this study is composed of and modified from the open source APM (Ardupilot Mega 2.6 autopilot) firmware and open source software Mission Planner, transmitted by ground-air XBee telemetry. In order to generate high resolution DSM (defines the terrain data including buildings and tree canopy) and mosaic orthorectified images, a total of 3767 photos were gathered by Sony ILCE-QX1 camera mounted on the crafts, during 5 fly missions, for a total coverage area of 33 km$^2$ with 8 cm Ground Sampling Distance (GSD). The coverage of the adjacent photos is kept at least for 85% overlap and 45% sidelap. The raw images are processed by Contextcapture and Pix4Dmapper software mutually, the data sets generated in this study are orthoimages and DSMs, with a grid spacing of 10 cm. Prior to the morphotectonic analysis (e.g. Deffontaines et al., 1991, 1993, 1997, 2016A; Pubellier et al. 1994), the quality of the data set (see whole data set on figure 2) needs to be evaluated. Eighteen ground control points are extracted with grid size of 2 m resolution of airborne Light Detection And Ranging (LiDAR) dataset and from the airborne LiDAR associated 25 cm resolution orthorectified image. Most of the ground control points are situated on crossroads, targeted and georeferenced from orthorectified image and elevation from the airborne LiDAR data, respectively. The comparison of the UAS DSM with airborne LiDAR data gives a root-mean-square deviation (RMSD) of 4.1 cm with maximum error of 42.5 cm from 26 sites of open bare ground area, e.g. roads, school playgrounds, unvegetated terrains, and parking lots. The elevation of the check points is averaged from an area of 4 m$^2$, equal to the grid size of airborne LiDAR data. The distribution of the ground control points and checked points are located on Fig 2b. Fig. 3 highlights the precision and resolution difference on the available topographic datasets. The 40 m grid DEM was generated in 1986 by an Aerial Survey Office from aerial photos, the 5m grid DEM was generated by several aerial photogrammetric agencies in 2004, whereas most of the aerial photos were acquired in 2003. The project of 5m DEM dataset was funded by the Taiwan Ministry of the Interior. On the other hand, the airborne LiDAR data was acquired in 2011 and used to generated a 2 m grid DTM (defines the geomorphologic elevation after removing the buildings, trees and vehicles) and DSM, but it is not authorized to publish even though it was generated by the authors. The ground control points used in this study were extracted from the airborne LiDAR dataset we acquired some years ago. Fig.3 highlights definitely visually the difference of quality, precision, and resolution of each digital topographic dataset. One may note the low quality of the 40 m DEM contrasting to the precision and resolution of our UAS-DSM. The Fig. 4 illustrates some case examples of the quality of both

the UAS-DSM as well as the orthorectified image of the Hengchun Fault. The morphostructures analyses based on photo interpretation are conducted accordingly (2 last columns).

## 3 Geology of the Hengchun area

### 3.1 Hengchun geological state-of-the-art

From a geologic and geodynamic point of view, the Hengchun Peninsula is interpreted as the northern tip of the Manila accretionnary prism (Malavieille et al., 2002; Chang et al., 2003). It is mainly composed of several lithological formations that are described below from the older to the younger ones (see their location on the cross-section of Fig. 1c).

The Mutan Formation is farther the larger outcropping formation of the Hengchun Peninsula. Middle to late Miocene in age, it is composed of folded classical turbidites made of shales interbedded with sandstones and conglomerates (channels and

levees) (Sung and Wang, 1986; Sung, 1991). The Mutan Formation is a very thick pile (locally several thousands of meters) and highly deformed turbidites (locally even overturned: e.g. CGS Geological map, Sung 1991; Chang et al., 2003).

The soft complex Maanshan formation, late Miocene, Pliocene up to Quaternary in age, is a mix of submarine erosion and depressions fill-in of both muddy Mutan and Kenting formations (Cheng and Haung, 1975; Page and Lan, 1983; Lin and Wang, 2001; Huang, 2006). The Maanshan formation is folded and outcrops partly in the South of the Hengchun Basin

(CGS geological map, Sung, 1991), see panorama of the Hengchun Basin on Figure 5. We believe that the Maanshan formation composes partly the basement of the Hengchun Basin and the offshore Manila accretionnary prism to the south (Lundberg et al., 1992 and 1997; Reed, et al., 1992).

The Kenting Mélange, which is only outcropping east of the Hengchun Basin, is known as a tectonic Mélange (e.g. Tsan, 1974; Page and Lan, 1983; Huang, 1984; Pelletier and Stephan, 1986; Huang et al., 1997; Chang et al., 2003, 2009A and B;

Deffontaines et al., 2016B, Zhang et al., 2016; Malavieille et al., 2016). It is interpreted as a chaotic tectonostratigraphic unit containing blocks of various origin (ophiolitic and sedimentary), size and lithology embedded in a sedimentary matrix (Malavieille et al., 2016). It is herein interpreted as a large tectonic breccia of composite ages that ranges from 1 to 10Ma (see Malavieille et al., 2016) as it mixes various sizes of different blocks and different turbiditic and shale bodies (Page and Lan, 1983; Lin and Wang, 2001, Chang et al., 2003). To our point of view this shaly/blocky tectonic Mélange is injected

(intruded) within the major fault zones south of Taiwan due to both the plate convergence and high pressure of fluids at depth (see also Deffontaines et al., 2016B). Future scientific work is needed as there is still different incompatible assumptions and hypotheses in order to both precise the processes that creates this Kenting tectonic Melange and to reconstruct (or inverse) its evolution through time (see synthesis in Malavieille et al., 2016).

The slightly tilted Hengchun Limestone are overlying the unconformable Maanshan formation on the Western side of the

Hengchun Valley, and overlying the Kenting Melange on the eastern side of the Hengchun Fault. As the Hengchun Limestone are folded differently and continuously on both sides of the Hengchun fault, it is unfortunately difficult to use the

existed dating results to estimate a precise vertical uplift rate in the absence of the precise location of sampling (Chen and Liu, 1993).

## 3.2 Updated Hengchun geology and neotectonics

Geological mapping in the fields lead us to look for active tectonic cracks within anthropic concrete dykes. They reveal the

clear lateral transpressive and thrusting component of the northern part of the Hengchun Fault (Figure 6). As the concrete dykes are manmade and very recent, it is easy to reconstruct and inverse the deformation following the Angelier's methodology settled along the Longitudinal Valley Fault (see also Lee et al., 1998, 2000, 2001, 2003, 2005, 2006).

From a structural point of view, the Hengchun Fault geological map that we propose (Fig. 7), is the result of (1) field studies (Figs. 5 and 6 taken January the 3rd, 2007), (2) the synthesis of previous geological mappings (Rokkaku and Makiyama,

1934 ; CPC, 1993; the CGS, Sung, 1991; Chang et al., 2003; Chen et al., 2005, and more recently the Giletycz et al., 2015) ; and (3) a detailed photo-interpretation of both the orthorectified mosaic of the UAS aerial photograph as well as the hill-shading of the high resolution UAS derived DSM (Fig. 7). The detailed structural photo-interpretation of the Hengchun Fault area checked in the fields, takes into account the basic morphostructural principles (Deffontaines et al., 1991, 1993, 1997; Pubellier et al., 1994) such as the geometry of the drainage pattern (bayonet tracks, curves and alignments), the alignments

of small scarps present in the flat Hengchun Basin (Fig. 4e, f, g, h, and Fig. 7, index 5) that helps to get the morphotectonic features (Fig. 7, faults that affect the topography, index 2 for instance). We also added on the Hengchun Fault geological mapping, geomorphological objects such as paleo-surfaces (index 5 in Fig.7) which are easily recognized by smooth textures and structures above the Kenting Mélange. Those paleo-surfaces are slightly tilted, folded and uplifted east of the Hengchun Fault and are witnesses of the vertical uplift of the transpressive tectonic activity of the Hengchun Fault. Unfortunately, due

to the slight tilt, it remains uneasy to extract fold axes from those paleo-surfaces. Anyway, it should be interesting to date these paleo-surfaces in order to get the middle to long-term activity of the deformation. We select also the chevrons from the morphostructural analysis of the high resolution UAS DSM (index 6 in Fig. 7). Chevrons correspond to the top of harder rocks tilted structural surfaces within the Kenting Mélange East of the Hengchun Fault. They reveal opposite tilted strata, consequently the presence of folds (anticlines as well as synclines, index 7 in Fig. 7) close and parallel to the Hengchun

Fault zone. Some tectonically offset and eroded mud volcanoes are also identified and mapped herein (index 3, in Fig. 6), especially in the South Hengchun Basin and close to the Taipower Nuclear Power Plant N°3. Those situated to the south of the study area were already evidenced by Giletycz (2015). They appear on the UAS DSM as characteristic "elongated conic volcanic" shape relief above the flat marine and alluvial lowland terrace. Their western side are locally affected by landslides (Fig. 7). We interpret these mud volcanoes as the outcropping traces of the shale intrusions within the Hengchun Fault zone

very probably due to fluids overpressure at depth in relation to the intense transpressive tectonic stress (see also Deffontaines et al., 2016B).

The Hengchun Fault is recognized as active (Bonilla, 1976; Hsu and Chang, 1979; Lee, 1999; Lin et al., 2000; Deffontaines et al. 2001, Lin et al., 2009, Deffontaines et al. 2016B). Effectively, geological mapping in the field reveals many active cracks within anthropic concrete dykes. These cracks highlight the clear transpressive and thrusting component of the

Hengchun Fault (Figs. 5 and 6). One may note that we cannot recognize clearly in the topography the trace of the Hengchun Fault as a unique straight line thrusting (see Fig. 7) contrasting to the CGS geological mapping (Sung, 1991) or its sinusoidal shape on the CPC geological mapping (1993). We only highlights herein the Hengchun Fault Zone which is made of locally aligned segments of faults that affect the eastern part of the very recent Hengchun Basin deposits (cf. index 2 in Fig. 7). But remember that regional geological maps aim at both stratigraphy and structure where fault trace is putting at where maximum stratigraphic throw locates (or just approximately) which may have large differences with the active fault trace situated strictly at active lines where last rupture locates. Moreover on land, last fault rupture can only be identified by geomorphic features; not by drilling or geophysical prospecting.

From the dating of the deformation point of view, the vertical long-term slip rate has been settled from marine terraces study and estimated of 6.3 mm/yr at Haikou (N Hengchun valley, see Fig. 2) and 3.8~6.1 mm/yr at Nanwan (South of Hengchun Valley, see Chen and Liu, 1993). Other marine terrace studies from Chen et al. (2005) show similar long-term slip rate.

## 4 Inputs of PS-InSAR interferometry on the Hengchun Fault

In order to get the regional tectonic activity of the Hengchun Fault, we processed 13 radar images (ALOS-1 PALSAR images acquired in L-Band λ = 23.6 cm, average incidence angle= 34.3°) in ascending mode (flight direction N350°E) between January 17th, 2007 and January 28th, 2011 so providing 4 years of monitoring, through the STaMPS software (Hooper et al. 2007). A series of 12 interferograms has been generated, all with respect to a unique "Super Master" as a reference image situated in the mid-time series that minimize both spatial and temporal decorrelation (Pathier et al., 2003; Champenois 2011A, Champenois et al., 2011B; Champenois et al. 2012, and 2014, Fig. 8). One may note that their perpendicular baselines situated in between -1194 and 2417m is much below the critical limit where all interferometric coherence is lost (Hooper, 1999). We used the 40m ground resolution Taiwan DTM in order to remove both the topographic and orbital components of the interferometric phase. All interferograms shows an excellent coherence in the Hengchun plain as well as in the hilly area. The method identifies 20133 PS pixels with an average density on the whole Hengchun Peninsula of 120PS/km$^2$, see Fig.9) that are characterized by their phase stability over the whole period. PS analyses lead us to extract a map of mean LOS surface velocities for each PS and also to catch their displacements through time series. Note that this LOS data is not projected onto the vertical component. The PS-InSAR base (fixed GPS station: GS59, correspond to the black and white star see its location close to Checheng - N of the Hengchun valley, on Fig.1 and 9) is chosen as it presents a stable (to very low) continuous deformation during the monitoring time period. Nevertheless, this location might be subject to small continuous uplift or small continuous subsidence that may explains local discrepancies with the GPS average annual displacement. We compare our PS-InSAR results (Champenois, 2011A and B, 2014) to three fixed GPS stations (HENC, GS57, GS59, see fig.9) and two leveling lines monitored acquired in the same monitoring time period (see Fig.10). For each of the 3 fixed GPS stations (HENC, GS57, GS59), it has been calculated an average displacement projected into the radar LOS by taking into consideration the various local incidence angle along the distance axis (Hanssen, 2001).

The results (Fig. 9) show an impressive density of PS (in the surroundings of the Hengchun Fault more than 140 PS/km²) in this area where both human activity and luxuriant vegetation prevail. The PS map reveals a clear continuous difference in LOS velocities across the Hengchun Fault where the Hengchun valley (western side of the Hengchun Fault) is subsiding with values ranging from -5 to -10 mm/yr contrasting with the westernmost Hengchun ridge which is uplifting. This is evidence in Fig.9 by a rapid transition from blue/green to orange/red PS dots. Taking into account the the PALSAR ascending satellite orbit and the NNW-SSE Hengchun fault trending both geometries is close to the best configuration to detect and measure the Fault displacements (Champenois, 2011). Consequently, this Fig.9 lead us to draw with much more accurate details than Shyu et al. (2005) and/or Lin et al. (2009), a new place for this active Hengchun Fault within the Hengchun plain (Champenois, 2011). The eastern side of the Hengchun Fault is characterized by the rapid increasing uplift of the Kenting Mélange up to a maximum that seems to parallel the Hengchun Fault. Farther east the PS LOS velocities decrease with the distance to the Hengchun Fault so the east of the Hengchun peninsula appears to be slightly tilted. We observe in the Hengchun peninsula a similar dissymmetric active vertical behaviour as in the Coastal Range (Eastern Taiwan) : the western part of the relief is highly actively uplifting and folding contrasting to the eastern part characterized by a decreasing uplifting and which is slightly tilted to the east (see also Deffontaines et al., 2016A). We note in Fig. 10B and Fig. 11, the coherence in between the PS LOS velocities and the GPS measurements (coloured large dots that we put into the same LOS radar geometry). Consequently, those PS-InSAR results offer a precise mapping of the active interseismic splays of the Hengchun Fault and allow the quantification of the tectonic displacements.

Regarding the Kenting Fault, with the data used in this study we see no gradient of PS LOS velocity, nor do noticeable differential displacements on the GPS stations situated around the Kenting Fault (see Figure 9 and profiles of Fig.10A-10B). Moreover from the geological mapping analysis (Chang et al. 2003), the Kenting Fault has a sinusoidal planimetric geometry that follows more or less the topographic isocontour of the relief. So the Kenting Fault is a low dipping thrust fault. Consequently, this fault may be either locked at the surface, or an inherited inactive fault, or have a tectonic activity that we do not observe on this PS-InSAR dataset during the monitoring time period.

We note a great coherence in between the PS and the available GPS measurements (colored large dots on Fig. 9) provided on two stations. For comparison purposes, the average annual GPS displacements acquired in the InSAR monitoring time period have been compared with the radar LOS. We compared also these PS results with leveling data acquired during the same period of time (between March 2008 and April 2011, during 3 campaigns of measurement). Figure 10 allows comparing the leveling data, the mean LOS velocities of PS along the leveling lines, and GPS data for the two stations situated close to the leveling lines. One may note the agreement between all of them. The local heterogeneity of the displacements (e.g. small subsidence as well as small uplift) within the Hengchun valley which is highlighted by both the HENC GPS station that gives the absolute uplift displacement and the PS-InSAR data with various coloured dots that range from blue to orange situated on both side of the O value.

Moreover, in order to better characterize the Hengchun Fault activity, and quantify the LOS velocity variation on both sides of Hengchun Fault, we computed the LOS velocity offset across the fault, using 31 profiles perpendicular to the fault line.

Each 10km long PS profile is perpendicular to the Hengchun Fault and characterized by an equidistance of 400m. In addition, each PS data situated in their both side (200m vicinity) are projected on each PS profile. Each data is only represented on only one profile. The numbers of PS dots per profile vary in between 400 and 1200 (Champenois, 2011A). This offset characterizes the slip rate of the fault. We present two specific profiles in Fig. 11 (see location Fig. 9). The two PS profiles,

which run transverse to the Hengchun Fault, one north of the Hengchun valley close to the Checheng city, and the second one southward (close to Hengchun city), show LOS offset less than 10 ± 2.5 mm/yr. One may note the high amount of PS reflectors situated on the Checheng and Hengchun city that confirm that on both profiles we evidence a clearly jump in the PS magnitude that correspond to the aseismic displacement of the Hengchun Fault (Champenois, 2011A).

So thanks to a series of more than 30 profiles perpendicular to the fault, we computed the interseismic LOS velocity offset

along the fault, from North to South, with LOS values (from min: 4.9mm/yr to max: 10.2mm/yr) reaching an average 8 ± 2.5 mm/yr during the monitoring time period that reveals by 2 linear regressions calculated on both side of the Hengchun Fault the clear tectonic activity of the Hengchun Fault (Fig. 12). One may note the good coherence between slip rates derived from PS analysis and the GPS measurement (projected in LOS taking into account each local incidence angle along the distance axis) represented by the yellow triangle, (see Fig. 12). This good fit in between the fixed GPS stations validate our PS results.

This is a major result from this study as it demonstrates the along strike interseismic activity of the fault (Champenois, 2011A and B; Champenois et al. 2012, this study), and allows obtaining the average annual interseismic LOS displacement of the Hengchun Fault, and its spatial variation along the fault. Variations of the velocity offset along strike the Hengchun Fault needs to be carefully monitored and analysed in order to better characterize potential seismic hazards especially the low interseismic creeping areas where stress may accumulate (see Fig.12) and may correspond to partially locked part of the

fault source of future earthquake (close to Hengchun city for instance).

StaMPS gives also access to time series of displacement, from 2007 to 2011 (see Champenois, 2011A and B and 2012). We can thus derive at each image acquisition date the displacement related to the fault (a cumulated slip value), then allowing monitoring its slip variation through time. The time series of slip values (Fig. 13) does not show any clear seasonal accelerations or decelerations, it only expresses a rather linear behaviour of the fault during the monitoring time period.

Anyway this time series should be updated with new SAR images in order to get a longer monitoring period (Champenois, 2011A and B). It should help to know whether the Henghun Fault as a linear displacement through time (or not).

## 5 Discussion: up-dated Hengchun active tectonic model

Many different tectonic models have been proposed on the active Hengchun Fault zone (thrust, left-lateral strike-slip, transpression as well as transtension see Chang et al. 2003, 2009). But they miss both structural and active tectonic

arguments to better understand the Hengchun Fault (e.g. Liew and Lin, 1987; Central Geological Survey and CPC geological mappings -1991and 1993 respectively; Chen and Liu, 1993, 2000; Chang et al., 2003, 2009A and B; Chen et al. 2005; Vita-Finzi and Lin, 2005).

The simplest model that we propose herein (Fig. 14) that fits with the oblique NW trending Taiwan geodynamic convergence (Suppe, 1984), the E-W displacements measured by GPS data (Yu et al., 1997), the geological dataset and the geological mapping (CPC-1993 and CGS- Sung, 1991; Chang et al., 2003 and 2009, Giletycz et al., 2015), our detailed high resolution UAS DSM photo-interpretation (Fig. 7) and our interferometric 2009-2011 PS LOS displacements (Champenois, 2011A and 2011B, Champenois et al. 2014, Fig. 9), is given for the western part of the onshore Hengchun Peninsula (Fig. 14).

From all used geodetic point of view (GPS, levelings and interferometric), if we generalize the two leveling sections which is coherent to both the three fixed GPS stations as well as the PS-InSAR results, the western Hengchun Peninsula is slowly uplifting (less than +10 mm/yr) contrasting to the slow subsidence of the eastern part of the Hengchun Peninsula (0 to -2.5 mm/yr). So as we reveal a continuous active differential displacements across both sides of Hengchun Fault which is deduced from levelings, GPS and PS-InSAR, Hengchun Fault shows the existence of a clear interseismic creeping component during the monitoring time period, with an average value of 8 mm/yr along the LOS, confirmed by the leveling monitoring (see leveling line 1: 8mm offset - Fig.10b). The variation of displacements on both sides of the Hengchun Fault zone reveals also the progressive folding of the Hengchun Fault zone: the eastern hanging wall is progressively shortening and uplifting (as revealed by GS57 and the 2 leveling lines) contrasting to the progressive subsidence of the Hengchun Basin (footwall situated West of Hengchun Fault- see Figs. 10b and c and 11 prof1 and 2). These displacements results (Fig. 9, 10 and 11) combined with the structural photo-interpretation (Fig. 7) show that the fault is only locally outcropping and that active folding occurs, increasing progressively with time, stress and strain at depth.

Consequently due to the rectilinear structural geometry, the Hengchun Fault appears to act as an almost vertical active left lateral transpressive strike-slip fault with an active uplifting and folding (anticline) on its eastern hanging wall, contrasting to the active subsidence of the western footwall (which corresponds to the easternmost part of the dissymmetric Hengchun Basin, Fig.14). Moreover the Hengchun Fault is not outcropping along one continuous North to South small straight line but describe a rather large "Fault Zone" made of little active fault segments (see location mapped in details, Fig. 7). One may note that the thrusting and left lateral strike-slip motion has been deduced from both from the morphostructural analysis of the high resolution Digital Terrain Model (see fig. 4c, 4d, 4g and 4h) and the obliquity of the GPS arrows already evidenced in the bibliography (See blue arrows in Fig.1, 9 and Chang et al., 2003). Consequently, the figure 14 proposes a simple model where the Hengchun Fault acts as a N160°E trending transpressive left-lateral strike-slip fault zone. Its active hanging wall is overthrusting and is actively folding the topographic surface that tilts and folds the paleo-surfaces: the Hengchun Limestone, and marine terraces see Liew and Lin, (1987) and Chen and Liu, (1993). The Hengchun Basin (Hengchun Fault footwall - and west of it) under the transpression is folded as well, and is actively subsiding with a parallel to the fault syncline axis that helps for the lagoonal/marine deposition of the Pleistocene/Holocene in age Hengchun Basin (Chen et al., 1991).

So Hengchun Fault as many faults in central and southern Taiwan present both left-lateral strike-slip and thrusting components such as the Chelungpu Fault reactivated during the Chichi Earthquake (September 21[st], 1999) as well as any Taiwan Foothill transfer fault zones (e.g. Deffontaines et al., 1997).

Dealing with the amplitude and the components of the active Hengchun Fault deformation, the two leveling lines reveals the vertical offset of the Hengchun fault, consistent with interferometric displacement toward the LOS and give the following value of 8 ± 2 mm/yr that correspond to the vertical geodetic interseismic slip rate. One may note that this short term value is a bit different than the vertical long term slip rate deduced from the marine terraces datings 6.3 mm/yr at Haikou and 3.8~6.1 mm/yr at Nanwan (Liew and Lin, 1987 and Chen and Liu, 1993). From this slight difference in between the long term and the vertical geodetic slip rates, we may deduce the fact that the Hengchun Fault may be not only a creeping fault. But due to the folding component of the eastern side of the Hengchun Fault, it is needed to locate much precisely the in-situ dated samples (Chen and Liu, 1993) to reconstruct and inverse correctly the long term deformation history.

Dealing with the active horizontal slip rate of the western Hengchun peninsula, it has been already published by Chang et al, (2003) by comparisons of the directions and amplitude of the three fixed GPS stations. But one may note that these values take into account both Hengchun and Kenting faults without individualisation.

## 6 Conclusions and perspectives

First, the flights of UAS with images acquisition lead us capable to produce through photogrammetry a high resolution DSM of the Hengchun Fault area with a less than 7 cm ground resolution and less than 40 cm precision. However for some places, the texture may change rate quicker and higher than the image capture acquisition, like for example the lakes or water surfaces due to sunlight reflection for instance, which causes noises in DEMs. In addition, concerning the DEM photo-interpretation, especially for the active geological structures, the terrain is easily to be modified by human activity in Taiwan: cities and farmer land are rapidly growing in some areas concealing morphotectonic structures. Fortunately, the ortho-mosaic images help to recognize those artefacts. Overall, the autonomous UAS and well developed photogrammetry technique allow generating conveniently high detail topographic information, and help to carry on long term scale morphotectonic study. Integrating levelings, three fixed GPS stations and PS-InSAR processing that gives the short term active movement of the Hengchun Fault and high resolution DSM structural photo-interpretation acquired from UAS demonstrates longer term topographic deformations and anomalies, thus providing one of different tectonic deformation to locate, characterize and quantify the active tectonics features.

Then with these so useful high resolution topographic data, we undertook a classical geological and geomorphological photo-interpretation that lead us to up-date and refine the pre-existing CPC-1993 and CGS-1991 geological maps, especially on both side of the Hengchun Fault (see Fig.4 and 7). The latter lead us to precise the structural geometry of the Hengchun Fault in the Peninsula (Fig. 7). Moreover, the levelings as well as the PS-InSAR interferometric processing that we carried out over the Hengchun Peninsula from 2007 to 2011 with more than 12 coherent interferograms reveal clear interseismic

displacements of the Hengchun Fault. Moreover, our interferometric results which are in agreement with levelings and fixed GPS data, reveal the active sections and folding of the eastern hanging-wall of the Hengchun Fault of about $8 \pm 2$ mm/yr. Furthermore the footwall is also subject to a complementary active subsidence related to the development of an active syncline parallel to the Hengchun Fault. These displacements are a cumulative vector taking into account both planimetric and vertical components of the deformation. Our PS-InSAR results are fully coherent with both two leveling lines E-W trending across the Hengchun Fault and three continuous GPS stations situated on both sides of the Hengchun Fault. Consequently, we enlarge our interpretation due to the PS density (more than 140 PS/km$^2$) that offers a spatial density of measurements that is greatly higher than that the one offered by GPS, allowing us a global and whole coverage and mapping of the active Hengchun Fault deformations (Fig. 9).

From the active tectonics conclusions, we deduce that the Hengchun Fault characterized by a high dip-angle acts as an interseismic creeping large left-lateral strike-slip fault zone with a clear transpressive vertical component of 8 +/-2 mm/yr (levelings and LOS) associated with active foldings. We evidence that the Hengchun Fault present a differential tectonic behaviour from north to south: in its northern part, the Hengchun Fault is narrow and clearly outcrops below the marine terrace with a measurable offset (see Fig. 4d). That contrast with the central and southern part of the Hengchun Fault where it is wider or larger (see many prallel and colinear faults and cracks that offset the topography on Fig.7) and submitted to active folding and only locally the fault is partially locked (close to the Hengchun city, see Fig. 12). In the central and southern part, as the PS interferogram spectrum increase rather continuously from W to E, transverse and above the whole Hengchun Fault zone, it means that the Hengchun Fault is actively folding at depth. Moreover, the lack of clear along strike PS discontinuity across the Hengchun Fault zone proves that it is not clearly offsetting the topography (and outcropping) during the acquisition monitoring time period (2007-2011). This should sign the slow continuous increase of tectonic stress at depth which is a major clue for a future earthquake in that area.

Unfortunately, the active differential displacement on both side of the Kenting Fault is much more difficult to highlight with the present geodetic datasets and our PS-InSAR results due to (1) the low fault dip angle deduced from its mapping geometry, or (2) an inherited inactive fault, or (3) a locked fault at the surface, and also maybe due to (4) the ductile rheology of the underlying formation (Mutan clay turbidites - MT). Consequently, it is also needed to much better geologically map and monitor the Kenting Fault in order to better understand its geometry, its rheology and active tectonic behaviour.

Nevertheless, the shallow seismicity (Deffontaines et al., 2016B) confirms the high onshore shallow tectonic activity that hits the Hengchun Peninsula. The deep offshore major Hengchun earthquake (December 26th 2006, depth: 44 km, $M_L$=7.0) also participated to the active deformation of the Hengchun Peninsula.

Further datings ought to be first located in respect to the Hengchun Peninsula folding then undertaken in order to better characterize the ages of the terraces, paleo-surfaces and the different deposits of the Hengchun Basin in order to inverse and reconstruct its tectonic history (Liew and Lin, 1987; Chen and Liu, 1993 ).

Finally, due to the average Hengchun Fault along strike variations (around 8 +/-2 mm/yr displacement toward the LOS confirmed by levelings - Fig.12), it is definitely needed to better constrain the Hengchun peninsula active tectonics in order

to prevent major destructions and major failure in the near future as for example to prevent destructions to the so sensitive energy industries (e.g. the Taiwan Nuclear Power Plant N°3 situated along the southern tip of the Hengchun Fault).

## 7 Acknowledgements

We are grateful to Andy Hooper for having developed and permitting to use the 'StaMPS/MTI' algorithms. We thank also the Japan Aerospace Exploration Agency (JAXA) for the ALOS 1 data in the scope of PI 112-0001 project (project number: 5226904000 PI: Benoit Deffontaines. Erwan Pathier (UJF Grenoble) and Benedicte Fruneau (UPE) are also deeply thanked for their supports during the Johann Champenois PhD thesis. This project was partially supported by the LIA ADEPT (now D3E), N°536 French CNRS-NSC (now MOST), and Taiwan MOST 105-2116-M-027-003.

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

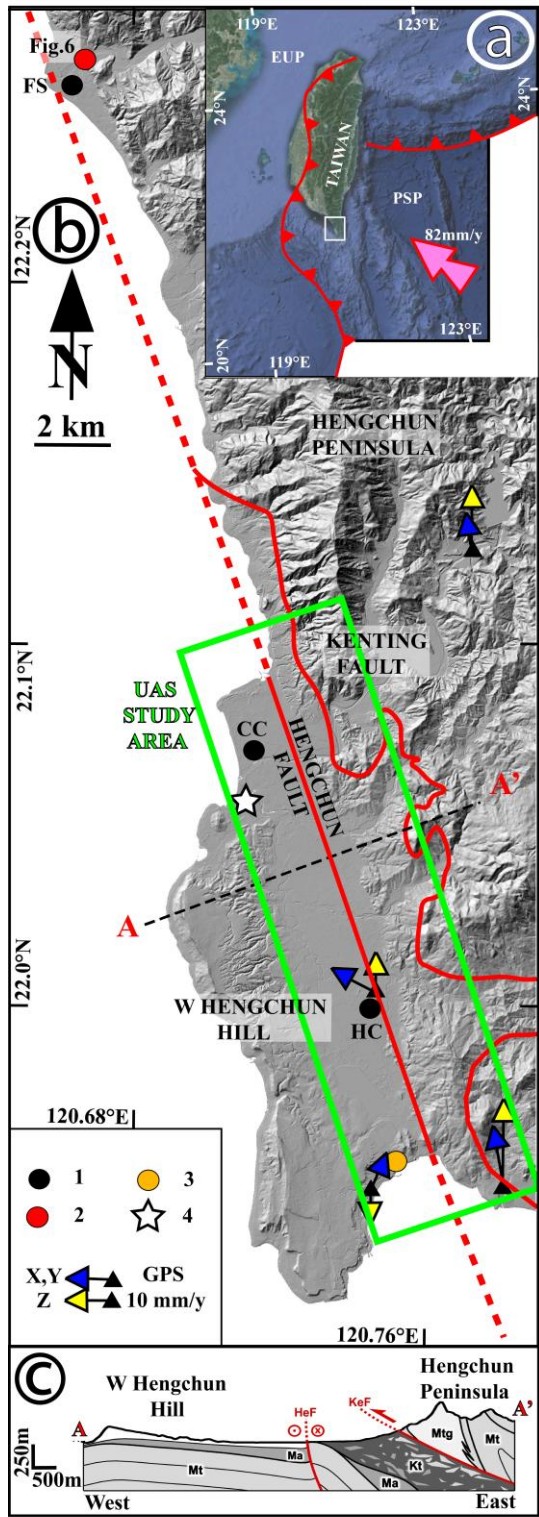

**Figure 1: Location of the Hengchun and Kenting Faults within Southern Taiwan. Fig. 1a: Location of the Hengchun Peninsula (modified from Google Earth, studied area: white quadrangle); EUP Eurasian Plate; PSP: Philippine Sea Plate. 82 mm/yr is the rate convergence of the Philippine Sea Plate toward Eurasia (Yu, et al. 1997). Fig.1b: W Hengchun Peninsula hill-shade from the 5m ground resolution Digital Terrain Model: The heavy red lines are Hengchun and Kenting Faults (modifed from Sung, 1991 - CGS geological map, and Chang et al., 2009B). Black Triangle: used GPS stations; Blue and yellow arrows represent horizontal and vertical GPS displacements respectively; Green quadrangle correspond to the UAS monitored area; A-A' location of Fig.1c representative geological cross-section; Fig. 1c: W-E trending synthetic geological cross-section (modified from Chang et al., 2009B; Zhang et al., 2016), Ma: Maanshan formation, Mt and Mtg: different facies of Mutan formation (Mtg: loshui sandstone - CGS geological map), HcL: Hengchun Limestone, Al: Alluvial deposits, KeM: Kenting Melange, HeF: Hengchun Fault, KeF: Kenting Fault. Black circle (1) : Cities (CC: Checheng, HC: Hengchun, FS: Fang-Shan); 2 Red circle (Location of photograph Fig. 6), (3) Orange circle Nuclear Power Plant N°3; (4) White star : GPS ground fixed station GS59 and base of this PS-InSAR dataset. The Fault displacements comes from Chang et al. (2003).**

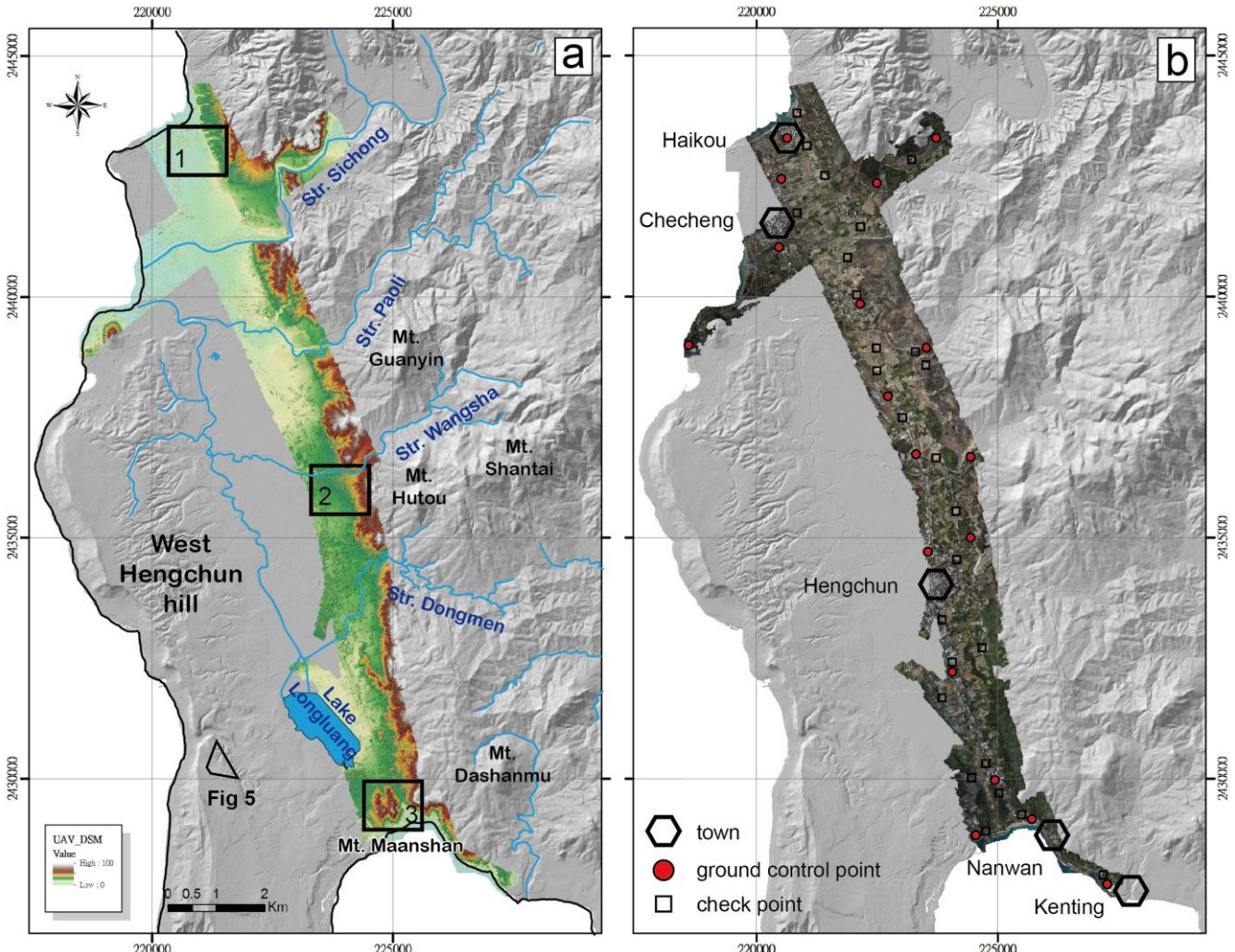

**Figure 2: UAS Digital Terrain Model (DTM) and mosaic of orthorectified images, indicate, Fig. 2a and Fig. 2b, respectively. The black hexagons, red circles and rectangles locate cities, ground control points and check points, respectively. One may notice the contrasting relief on both sides of the Hengchun valley: its western flank is a homogenous structural surface slightly dipping east contrasting to the highly eroded eastern part. For both Fig. 2a and 2b, the black and white hill-shade DTM correspond to a part of the 5m ground resolution Taiwan DTM.**

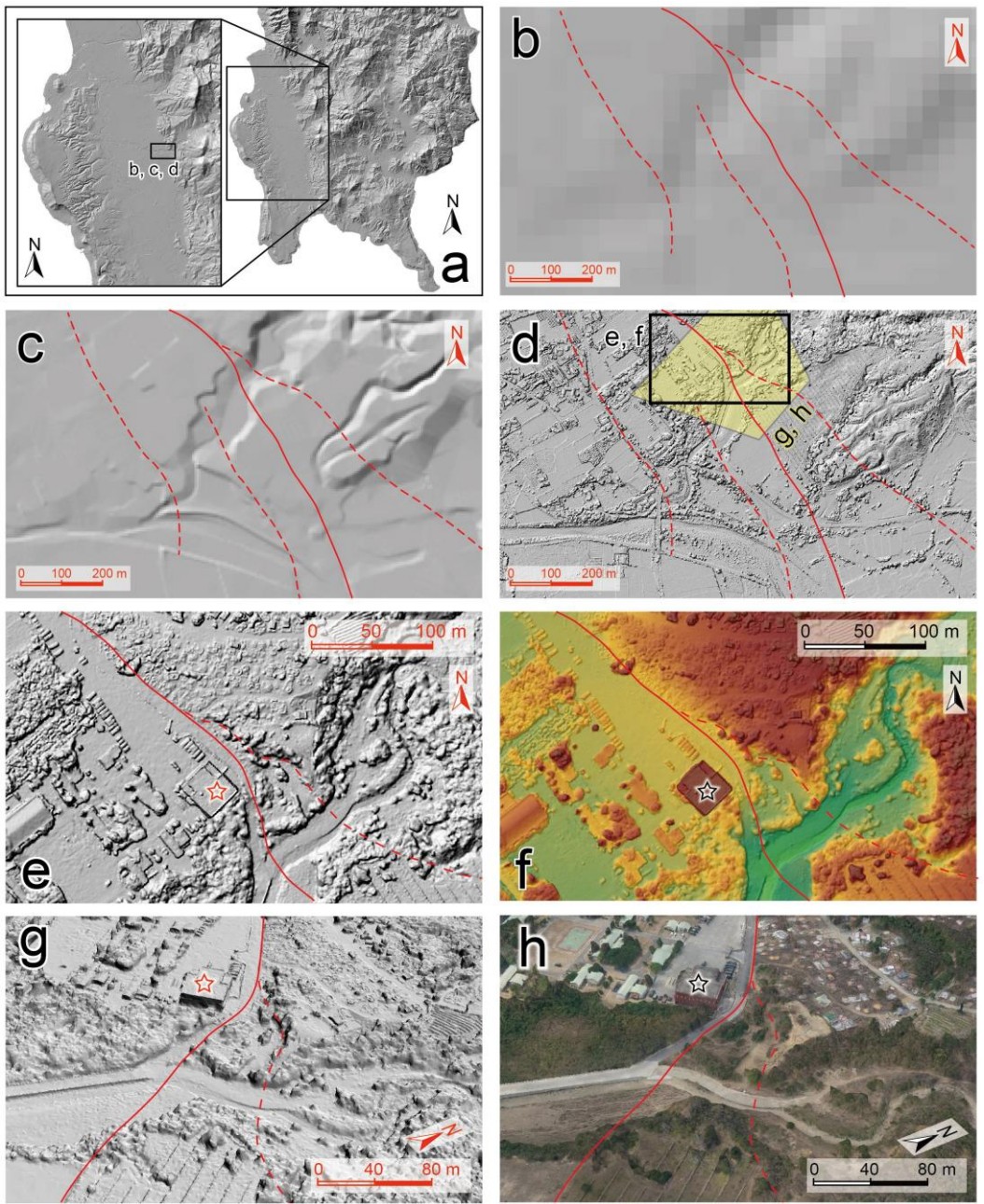

**Figure 3: Comparisons of Topographic Digital datasets for the same area acquired from different sources, including 40m DTM, 5m DEM and 0.13m UAS DSM. Heavy and dashed (inferred) red lines corresponds to fault lines. (a) Location of the dataset of fig b, c and d; (b) 40m resolution DTM commonly used in Taiwan; (c) 5m resolution DEM generated in 2003; (d) 0.13m resolution UAS-derived DSM generated in this study; One may see on the same area the great difference of quality of the different digital topography. e to f highlights different part of the UAS DSM : (e) enlargement of the hillshaded UAS DSM; (f) enlargement of color shaded UAS DSM; (g) side view of the hillshaded true 3D model; and (h) side view of the image draped onto the true 3D model (g). The side views of (g) and (h) are captured at the same position and orientation. The star indicates the same building shown on (f), (g) and (h). The location of the figs (b), (c) and (d) indicated on (a), whereas the figs (e) and (f) indicate on (d). One may see the quality of the newly acquired UAS DSM that enable us to photo-interprete so precisely the location of the geology and the faults.**

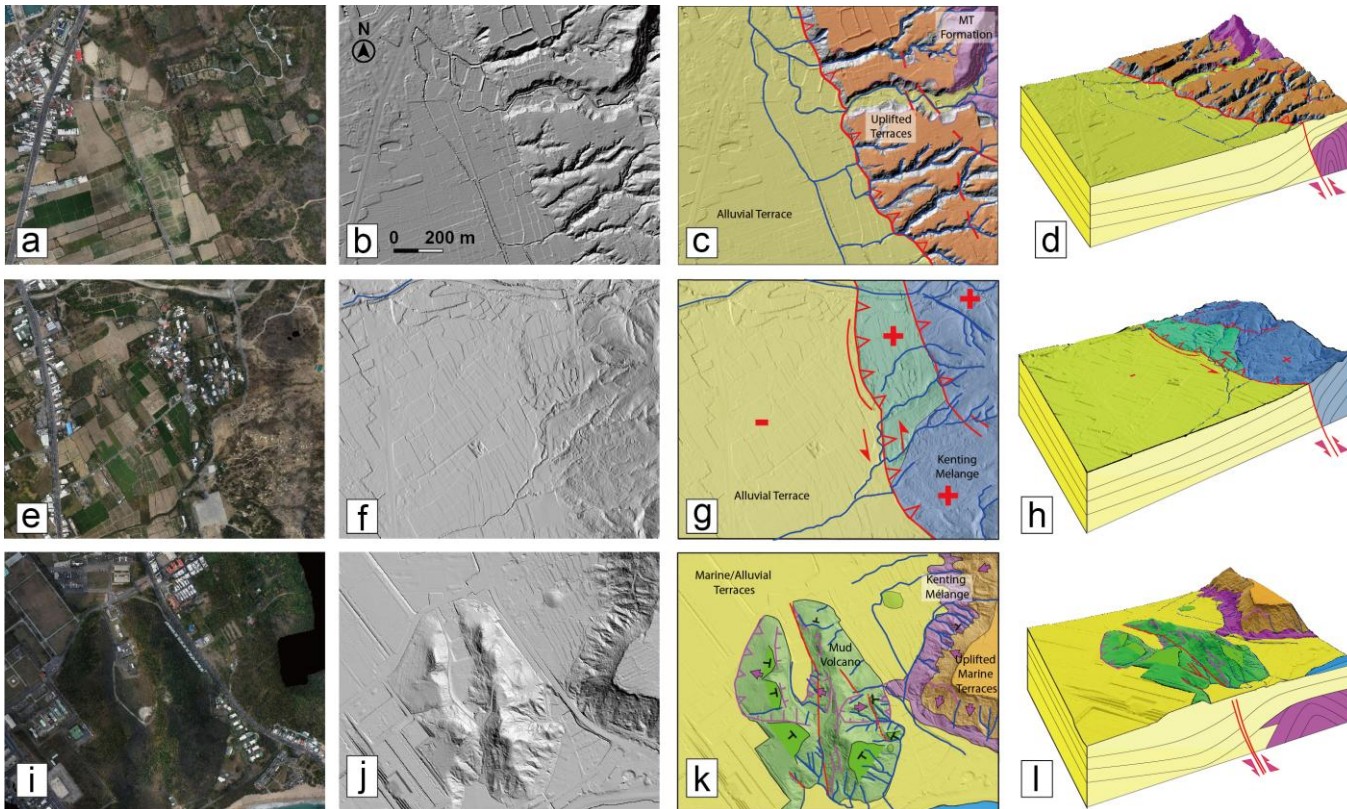

**Figure 4: Three geological photo-interpretation case examples taken from North to South of the Hengchun UAS Digital Elevation Model (DEM).** Each box cover the same area (1.3 km x 1.1 km): First column (a, e, i) : mosaic of the georeferenced aerial photographs (orthophoto) of each study area. Second column (b, f, j) : Black and white hill-shading relief from UAS derived DTM; Third column (c, g, k): geological map from the high resolution hill-shade DTM where heavy red line corresponds to the Hengchun Fault, blue line to the drainage, in light yellow : quaternary alluvial and marine terraces, in purple/ light blue due to the DTM transparency the Kenting Melange, in brown the uplifted marine terraces, in light green Mud-volcano and dark green its associated structural surface (chevrons with radial outward dips in black line), purple lines boundaries of landslides ; + : uplifted area; - : subsiding areas, NPP: Taiwan Nuclear Power Plant N°3, (see location Fig.1), Mud volcano; Fourth column (d, h, l) : 3D geological mapping with their associated geological cross-sections that shows the Hengchun Fault at depth (red arrows : active fault displacements). blue lines : rivers, The first line of images highlights the active tectonic scarp of the Hengchun Fault which differentiate 2 blocks: an uplifted (+) locally eroded terraces to the east and a flat lowland sedimentation area to the west. The second line of images highlights the left lateral offset geometry (red arrows) of tributaries (blue line) on a glacis which is uplifted (+) on the east of the Hengchun Fault. The third line of images shows a deformed N-S trending elongated mud volcano (MV - MaanShan) previously evidenced by Giletycz, 2015 in its PhD dissertation) situated close to the Taiwan Nuclear Power Plant N° 3 (NPP and see infrastructures on the lower left corner of the deformed mud volcano) and a flat uplifted (+) terraces (paleosurface of Fig. 6) on the western side of the captions.

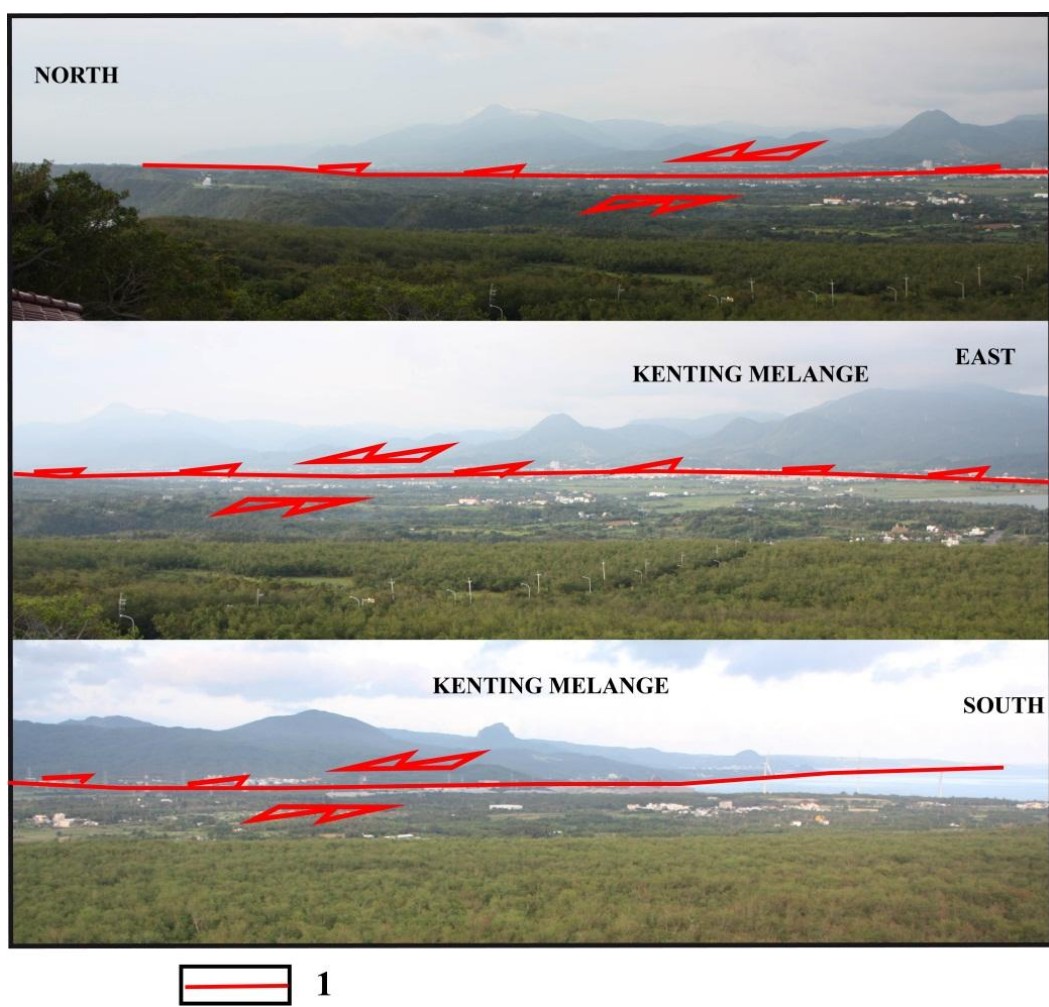

**Figure 5: Panorama of the Hengchun Fault observed from the W Hengchun Hill to the East. Red lines (1) correspond to the left lateral compressive Hengchun Fault. Note the isolated summits within the Kenting Melange that is composed of highly deformed shales with various huge blocks (olistoliths?) interpreted herein as a tectonic breccia. The location and the direction of the photos where it were taken are indicated on Fig 2a.**

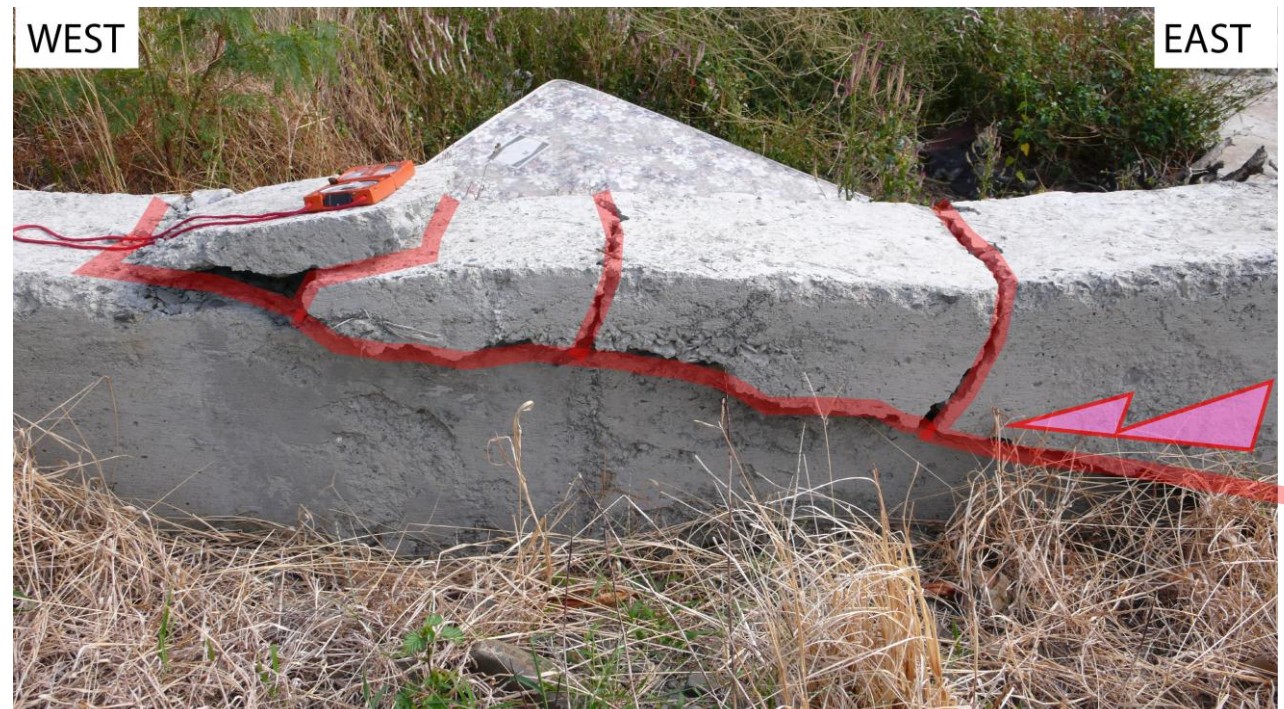

**Figure 6:** Field work photograph of compressive features affecting recent concrete dykes after the Hengchun earthquake (Dec. 26[th] 2006, $M_L$=7.0), see location on Fig. 1 near the Fang-Shan village (22.26°N, 120.66°E). The red compass gives the scale of the outcropping deformation (20cm). Thrust associated with three back-thrusts highlighting a small pop-up (below the compass). The red arrow reveals the thrusting component that affects the new concrete dykes.

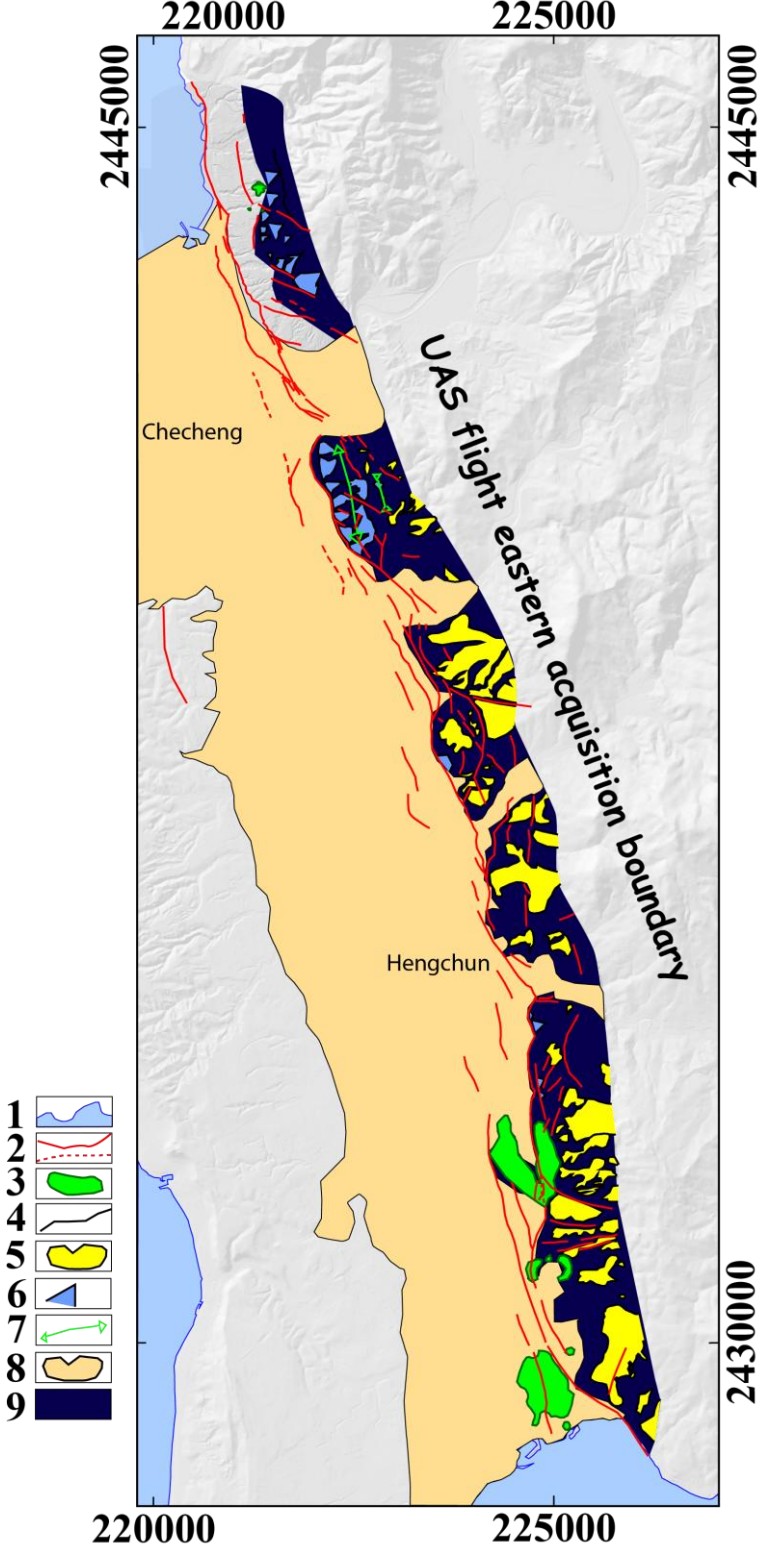

**Figure 7: Morphostructural map of the Hengchun Fault: 1. Shoreline; 2. Faults (red lines: certain faults, red dashed lines: inferred); 3: Mud-volcano; 4: Lithological boundaries; 5: Paleo-surfaces; 6: Chevron corresponding to tilted structural surface, 7: Fold axes; 8: Hengchun valley alluvial and marine deposits; 9: Kenting Mélange. Note that the Hengchun Fault is made of numerous small parallel faults affecting a rather wide area of the eastern Hengchun valley. It is associated with a N-S trending fold close to Checheng. Some folds that tilt paleosurfaces and NE-SW, E-W and NW-SE oblique faults affect the Kenting Mélange.**
**The western extension of the Hengchun alluvial plain (8) is deduced from the 5m DTM.**

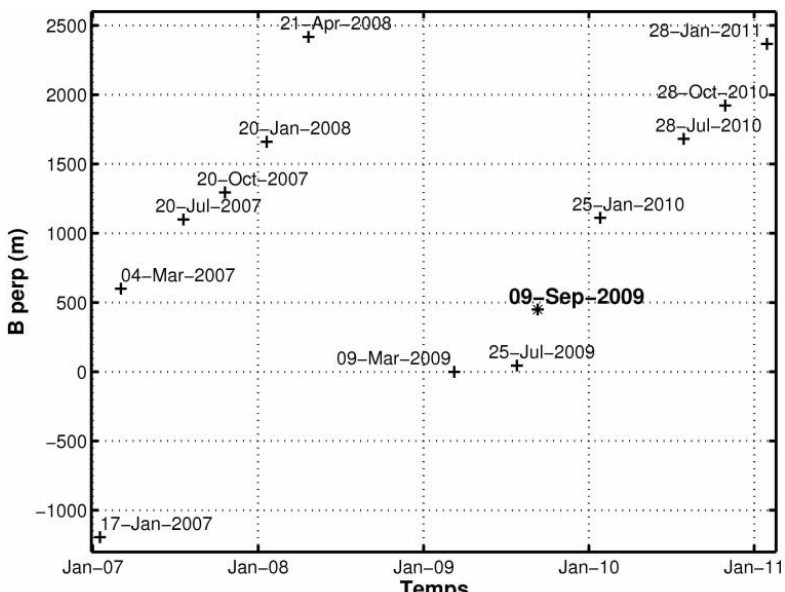

**Figure 8: Available ALOS-1 SAR images and their associated perpendicular baselines. All slave images are linked to the best master image chosen in the middle of the time series that maximizes the sum correlation of all interrerograms (Hooper et al., 2007).**

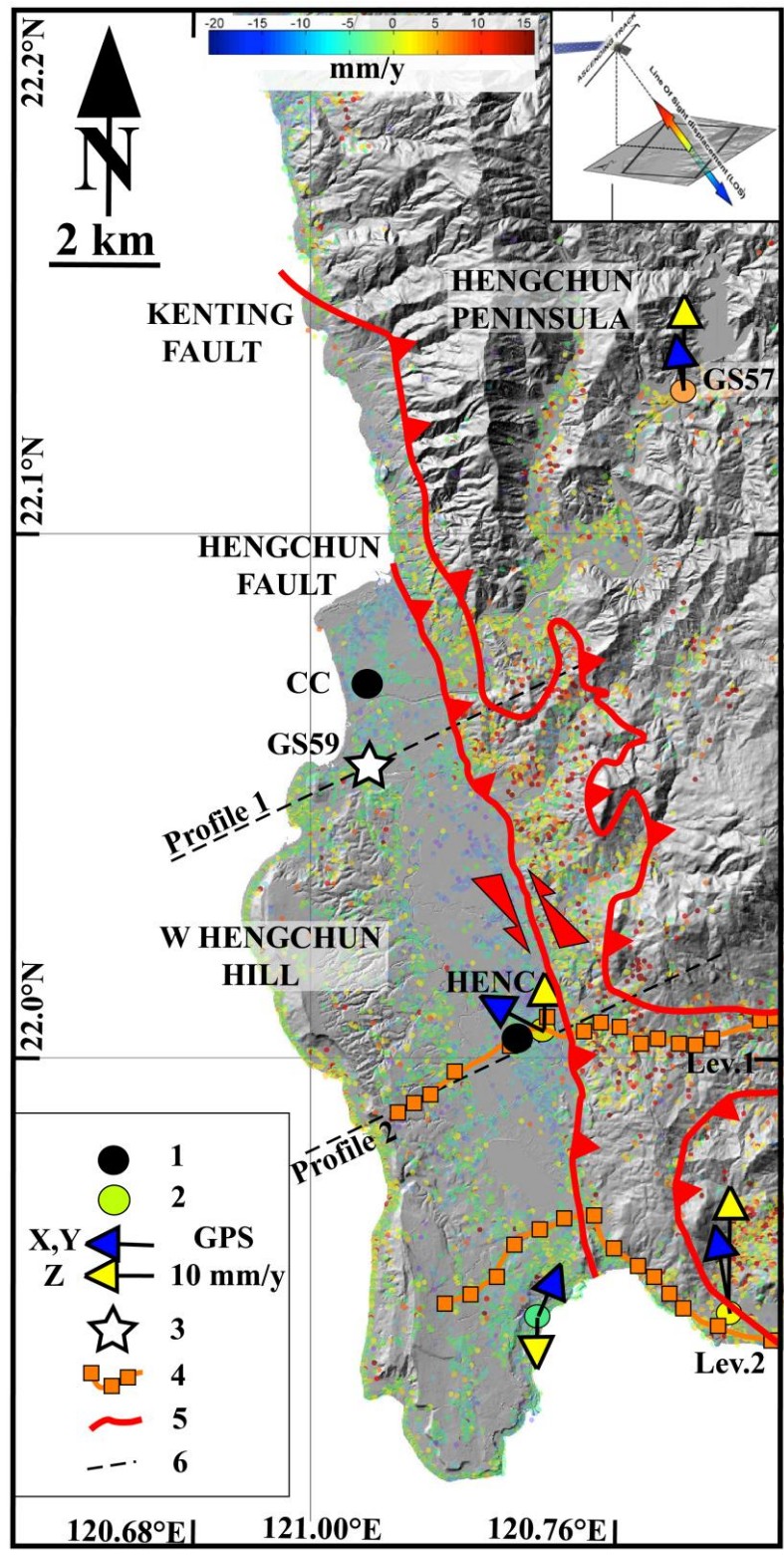

-20 -15 -10 -5 0 5 10 15
mm/y

KENTING
FAULT

HENGCHUN
PENINSULA

GS57

HENGCHUN
FAULT

CC

GS59

Profile 1

W HENGCHUN
HILL

HENC

Lev.1

Profile 2

Lev.2

2
X,Y    GPS
Z      10 mm/y
3
4
5

22.2°N
22.1°N
22.0°N

120.68°E    121.00°E    120.76°E

**Figure 9: Mean LOS velocities of PS-inSAR between Jan. 2007 and Jan. 2011, superimposed to the shaded 40m ground resolution S. Taiwan Digital Terrain Model. The large black circles are the used GPS points and the color in the circles indicates their average LOS velocity component during the monitoring time period. The black and white star (situated close to Haikou - see location on Fig. 1 and 2) correspond to the GPS ground fixed station GS59 and is the base of this PS-InSAR dataset. Black and grey arrows give their horizontal and vertical velocity. Red line: Hengchun Fault. Location of profiles 1 and 2 (see Fig. 11). One may note the active surrection and folding east of the Hengchun Fault contrasting with the subsidence of its western part (Hengchun valley).**

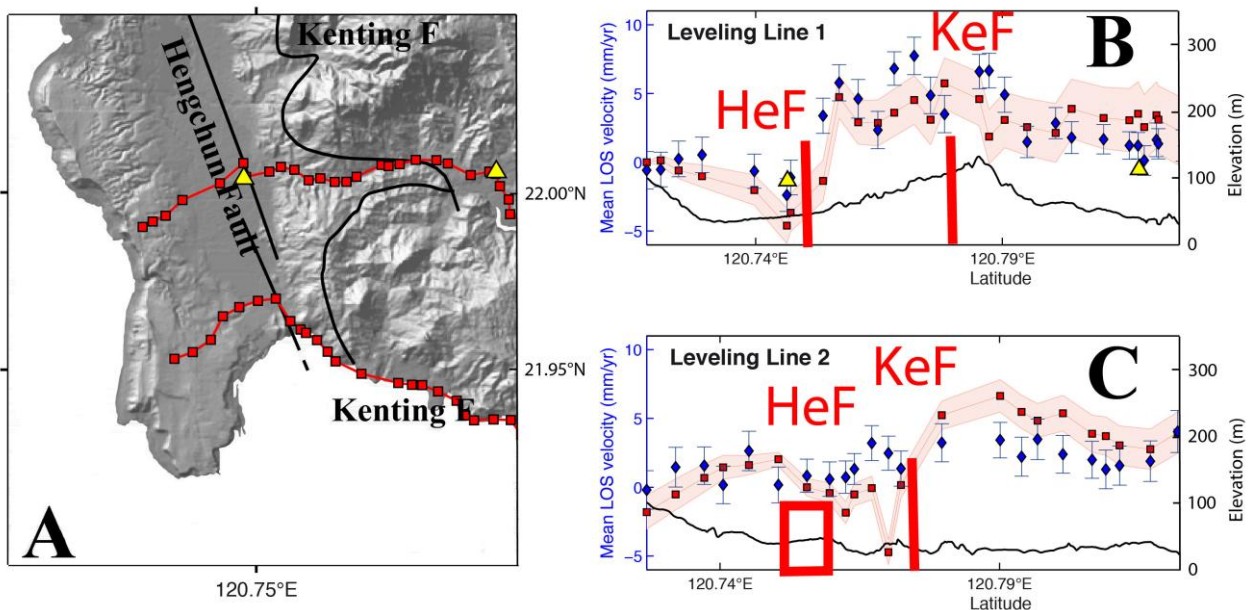

**Figure 10a: Location of the leveling lines on the Hengchun Peninsula. Fig. 10b and 10c: associated PS mean velocity with error bars (blue dots) leveling data (red square and in pink: error bar) and GPS stations data (yellow triangle), projected on LOS. Levelings highlight the vertical component of the deformation. The easternmost and westernmost reference points are on an enclosed curve line of the Class 1 level reference net defined by the country's Satellite Survey Center of Department of Land Administration (see SSCDLA). As leveling line 2 parallels the Hengchun Fault (points 6 to 8) it is represented as a red quadrangle. Note the general agreement of both interseismic leveling and PSI vertical topographic displacements on those two transverse profiles across the whole Southern Hengchun Peninsula. They show subsidence in the western part of the Hengchun Fault (HeF) contrasting with uplifting due to active folding on its eastern side. In addition, there is few significative vertical interseismic displacement on the Kenting Fault location on both profiles during the monitoring time-period.**

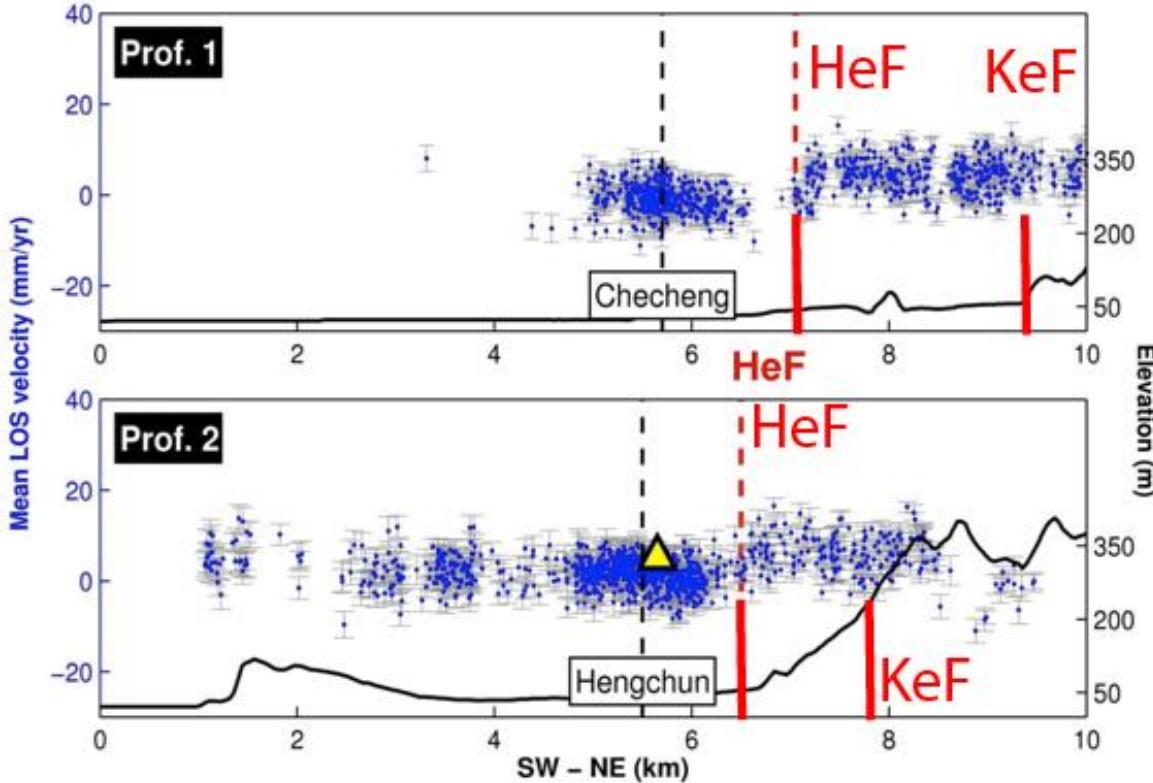

**Figure 11: E-W projection of mean LOS velocity in mm/y (blue dots with error bars, E on the right) on 2 profiles transverse to the northern part of the Hengchun Valley (close to Checheng and to Hengchun city, see profile location on Fig. 9), Black line: topography, Yellow triangle: GPS point, HeF: Hengchun Fault, KeF: Kenting Fault. Note that both figures 10 and 11 shows similar topographic displacements : on both profiles the LOS offsets close to 10 ± 2.5 mm/yr associated with the Kenting Mélange uplift east of the Hengchun Fault which contrast with the relatively subsiding Hengchun valley. Kenting Fault has no vertical interseismic offset during the monitoring time period.**

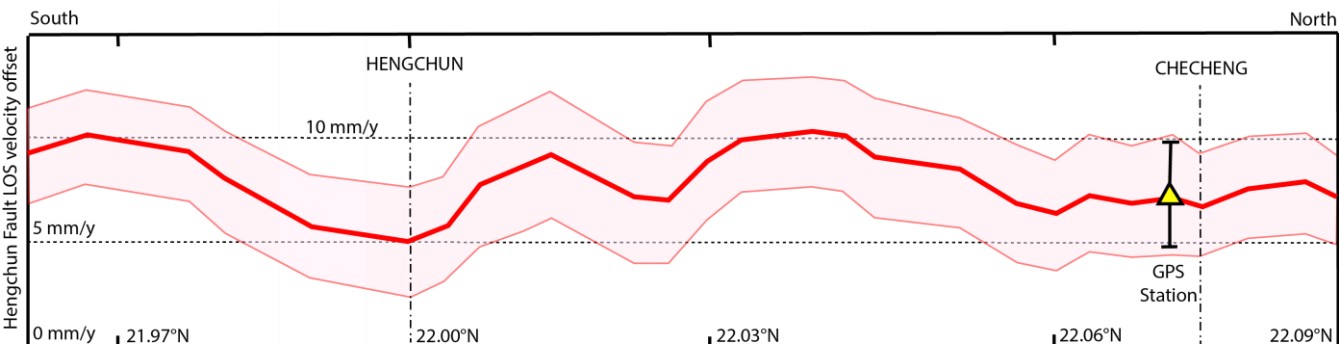

**Figure 12: Hengchun Fault along strike variation of LOS velocity offset (perpendicular to the Fault, in mm/yr) including Hengchun and Checheng cities. Note the active interseismic displacement and its along strike variations which confirms that Hengchun Fault is an active interseismic creeping Fault of Taiwan. Furthermore the LOS velocities variations are so important in terms of Seismic Hazards as they reveal the location of slowly creeping areas where stress may accumulate at depth such as those areas close to both Checheng and Hengchun cities, and higher active tectonic areas such as in between Checheng and Hengchun or south of the Hengchun Valley. Part of the deformation may also be distributed in the eastern part of the Hengchun Fault (e.g. the Kenting Fault and/or within the intrusion of the Kenting Melange in between both He and Ke faults see Fig.14 below).**

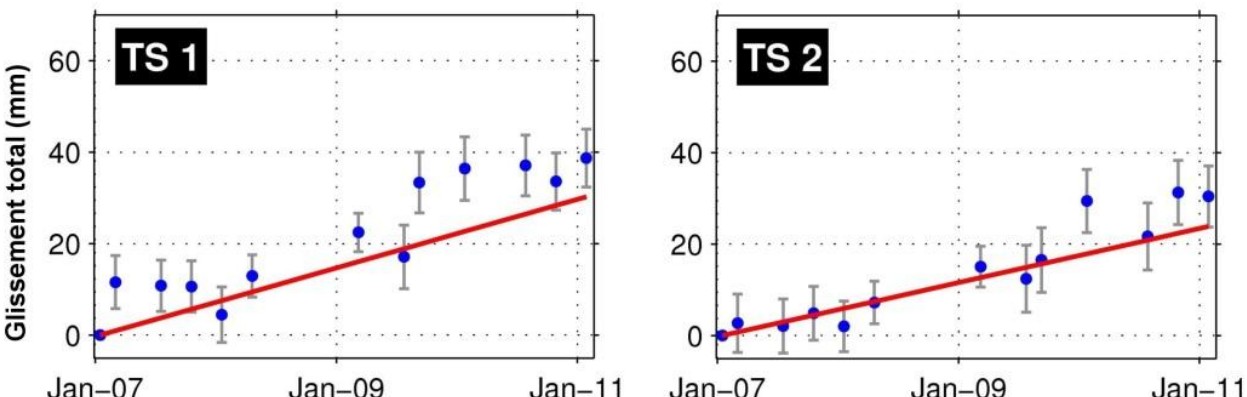

**Figure 13: Time series of cumulated slip (on profiles 1 and 2) where no particular seasonal effects prevail.**

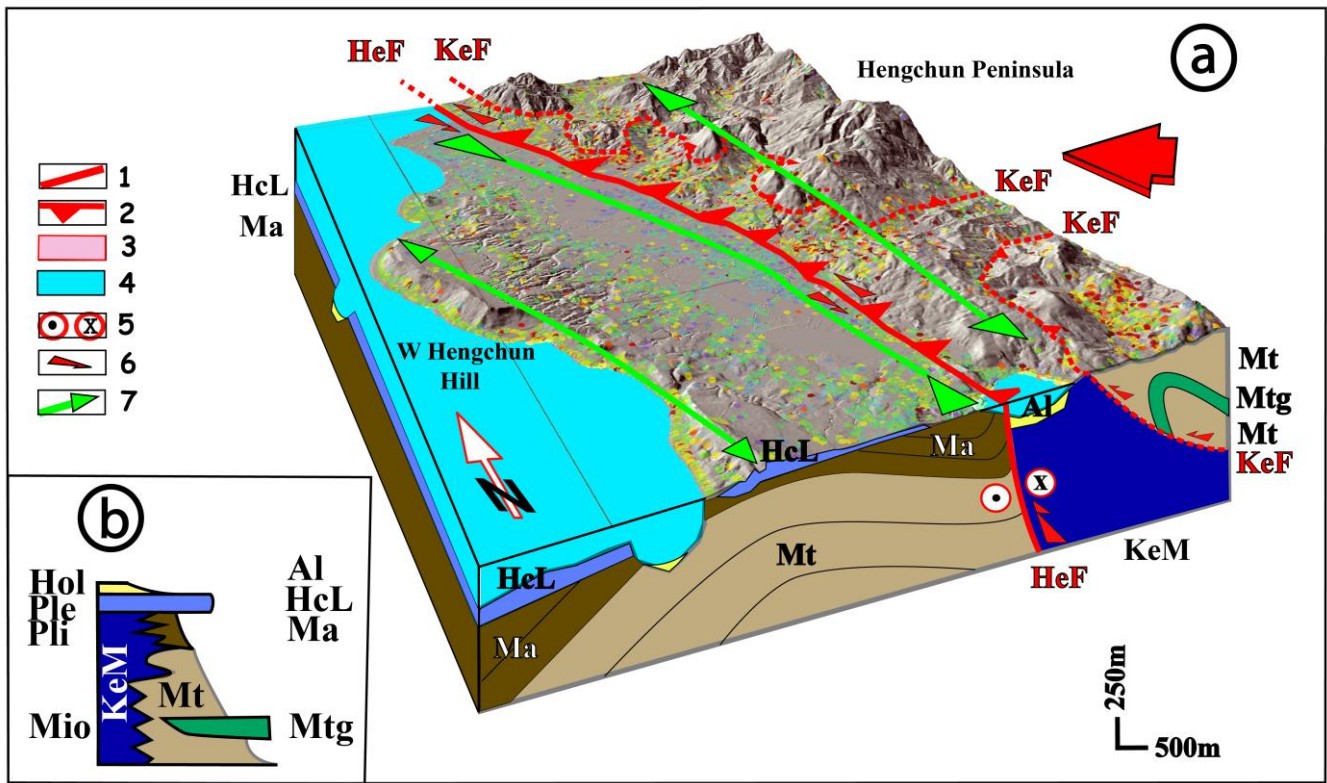

**Figure 14: (a) Simplest model of active interseismic tectonic deformation of the Hengchun Fault during the monitoring time period. It is compatible with the oblique geodynamic convergence (large red arrow on the right), GPS, leveling, and interferometric active displacements, and the detailed UAS DSM photo-interpretation. In addition, it is compatible with Rokkaku and Makiyama, 1934, CGS, CPC geological mappings, Chang C.P. et al., 2009B, and Deffontaines et al. 2016B. 1: Fault ; 2: Thrust ; 3: Fault plane ; 4: Taiwan Strait / South China Sea ; 5: Left lateral component ; 6: Transpressive left lateral component ; 7: Fold (anticline and syncline). The GPS data and PS mean LOS velocities implies that Hengchun Fault acts as a left-lateral transpressive strike-slip fault with an actively folding and uplifting anticline (hanging wall) contrasting with the aligned active subsidence of the Hengchun valley that act as a trending parallel to the Hengchun Fault. One may see the Hongtsai canyon at sea, see Deffontaines et al. (2016B). (b): Simplified Lithostratigraphic column of the Hengchun formations in the W Hengchun Peninsula : Al: Alluvial deposits, HcL: Hengchun Limestone, Ma: Maanshan formation, Mt and Mtg: different facies of Mutan formation (Mtg: loshui sandstone - CGS geological map), KeM: Kenting Melange ; HeF: Hengchun Fault, KeF: Kenting Fault; Mio: Miocene; Pli: Pliocene; Ple: Pleistocene; Hol: Holocene strata (modified from CGS, CPC, and pre-existing references).**