# Peer review of "Active tectonics of the onshore Hengchun Fault using UAS DSM combined with ALOS PS-InSAR time series (Southern Taiwan)"

_Natural Hazards and Earth System Sciences, 2017_

## Referee Comment (RC1) · Anonymous Referee #1 · 3 Mar 2017

The comment was uploaded in the form of a supplement:
http://www.nat-hazards-earth-syst-sci-discuss.net/nhess-2017-55/nhess-2017-55-RC1-supplement.pdf

---

## Referee Comment (RC2) · Anonymous Referee #2 · 15 Mar 2017

This paper characterizes the Hengchun and Kenting faults in southern Taiwan using a very high resolution digital elevation model acquired by unmanned aerial vehicles and new PSI interferometric data obtained from ALOS images. Their work emphasizes that the two active faults were not precisely known before in terms of its locations and regional tectonics. They carried out this work to better characterize the segments of an accretionary wedge-related active faults and hopefully to update the current understandings of these active faults on land.

I think the paper is generally clearly written, though some typographic or grammar errors can be found here and there, which should be easily corrected. Their work has also demonstrated that the new findings from the high resolution topography and

[Figure]

ALOS images are indeed necessary to update the active faults that may potentially lead to significant earthquakes in the region, where a nuclear power plant is present nearby.

In view of its new contribution to the characterization of the Hengchun and Kenting faults in Taiwan and the proposed potential roles that they may play in the accretionary wedge, the work should be worthy of publication following a minor revision to perfect its presentations.

Here are some comments and suggestions for the authors.

1. The authors can elaborate on the section on high resolution digital terrain model acquired form UAS, which is an important part of the work and deserves to be more detailed. This addition can be accompanied by revision of Figure 3 (see comment #6) to clearly present the information of the UAS derived images.

2. Page 7, line 6-8: It may not be appropriate to state that the Hengchun Fault is not only a creeping fault by referring to work in progress. Please either present more of the data and reasoning, or leave out this inference for not jumping to conclusions.

3. The section on conclusions and perspectives is inappropriately long, and actually, some of the content is most suitable for the discussion section. Please leave the conclusions section for the most important findings of the work, that is, the UAS mapping result, PSinSAR result, and tectonic interpretations for clarity and ease of reading for the readers.

4. Figure 1: The abbreviated words HeF and KeF need to be indicated in the caption text. Inconsistent wordings are present, for example, Hill and hill in the figure, Fig.1C and Fig.1c in the caption.

5. Figure 2: Please label some local topographic features in the blank figures for clarity.

6. Figure 3: The caption text of Figure 3 is unclear and should be rewritten to state more clearly what these images represent in an orderly, straight forward manner. The

images should also be labeled with words for clarity as to avoid misreading the true intention to show. Where are the active tectonic scarps? Where are the left lateral offset of tributaries on a glacis? Where is the deformed elongated mud diapir? And where is the nuclear power plant? These are not well indicated in the images and a clear presentation of these features is important for readers.

7. Figure 4: The legend at the bottom of the figure is not clear and can be removed. Instead, please label these panorama views with simple text, so that the readers will understand what features you are pointing to.

8. Figure 5: . . . red compass "gives" the scale . . .; . . . red arrow "reveals" the thrusting . . .

9. Figure 6: Labeling some local topographic features in the morphostructural map can be helpful for readers to understanding the significance of the map. Also, the colors in the map seem inconsistent with the legend examples.

10. Figure 10: The caption says that one may note the general coherence of the topographic displacements. But the coherence is not clear based only on the selected profiles that are drawn on the diagram. One possible way to reveal the coherent topography would be using a swath profile approach to represent a wider zone of the topography. Otherwise, it is hardly clear to conclude the coherence in the diagram.

11. Figure 11 & 12: Clear legends for the figures are needed.

12. Figure 13: This should be the most important contribution of the work. However, the details in the figure are not clear enough. Also, the pink area on the lower right corner of the figure is strange and may not show up what the authors might want to indicate in the caption. A revision of the figure is strongly advised by adding more information presented in the paper, including minor lineaments, labeling meaning texts, and a more realistic view of the block diagram to conclude the paper's contribution.

---

## Author Comment (AC2) · 12 Apr 2017

Reviewer 2 : modifications done in a comment. Thanks for the general overview of our paper. We answer below to each suggestion:

1. We have modified the Fig.3 adding the structural interpretation on it in order to be more informative and clearer for the interest of UAS DTM photo-interpretation. Moreover we precise in the text the treatments that had been processes. 2. We precise the PS short term deformation slip rate misfit with the long term slip rate (deduced from the elevation of dated marine terraces). We may interprets this (little) difference by potentially an earthquake behavior but further work (that are in progress) is needed to go farther on that point. 3. We agree that the conclusions and perspectives are long

BUT we believe that it is important to deal in the conclusions with both the technical and methodological point of view AND the active tectonics point of view. So we precise in the text these both aspects that conclude completely this manuscript. 4. Figure 1: HeF: Hengchun Fault, KeF: Kenting Fault is added in the text. We homogenize hill, and Fig. 1c on Figure 1. 5. Figure 2: The figure is been improved. 6. The text of the different Figure captions (especially for Fig.3) is completed and clarified. 7. Figure 4 : The figure caption of Fig. 4 is modified. This figure is important as it shows to the reader the relief of the studied area (Kenting Mélange and so on). 8. Figure 5 : gives the scale (done) 9. Figure 6 : some label had been given (Checheng, Hengchun...)+ transparencies on the colors had been removed so the colors fits now with the legend. 10. We modified the text of Fig. 10 in the legend and in the text for the coherence of PS-INSAR and GPS data. As we added the error bars on PS-InSAR data, the mean velocities explain this apparent large distribution. This coherence is also seen on Fig. 11. 11. We modified the legend of Fig. 11 and 12. 12. Figure 13: This figure had been deeply modified (remove of the pink area in the lower right corner; we added the relief in 3D).

We have done all suggested modifications suggested by both reviewers, where the changes had been highlighted in blue, please see the attached file: Hengchun\_nhess\_0412.pdf.

Please also note the supplement to this comment: http://www.nat-hazards-earth-syst-sci-discuss.net/nhess-2017-55/nhess-2017-55-AC2-supplement.pdf

**Supplement:**

[revised manuscript text omitted]

---

## Author Comment (AC1)

**Active tectonics of the onshore Hengchun Fault using UAS DTM combined with ALOS PS-InSAR time series (Southern Taiwan)**

Benoit Deffontaines1, 2, Kuo-Jen Chang2, 3, Johann Champenois4, Kuan-Chuan Lin5, Chyi-Tyi Lee2, 6, Rou-Fei Chen2, 7, Jyr-Ching Hu2, 5, Bénédicte Fruneau2, 8

[revised manuscript text omitted]

---

## Author Response (AR1)

**Letter to the editor and reviewers**

Thanks to have taken into consideration our manuscript. Please find below our responses and modifications that we have done following the interesting reviewers suggestions.

*1) The structure of the paper needs to be reorganized. The current sections 2, 3, and 4 are provided according to the methodological topics (UAS, field mapping, and InSAR), but each is the mixture of methods and results (and sometimes general facts by previous studies). The method and result would be better described separately for the clarity of the paper. I would recommend making separate sections of Methods, and Results, under which the individual topics (UAS, field mapping, InSAR etc.) are provided. The general geology and geomorphology (that are not derived from this study) can be described in Study area before Methods.*

We understand this suggestion to modify the plan of our paper. But the methodology in this case will deals with very different techniques without commun points: DEM acquisition and processing, morphostructural interpretation as well as INSAR processing. As all these methodologies are so different, we may have a mixed in the methodologies.

For the plan of our paper and in order to be clearer, we propose to follow the "classical geological way" which is to focus first on the location of active faults by the high resolution of UAS DEM processing, and using morphostructural interpretation then to focus with a different paragraph on the characterization followed by the quantification of the active fault displacements toward the LOS by using independant INSAR results. We did not insist that much on the INSAR processing as it is already given by other authors. Moreover, using our plan we avoid many repetition.

*2) The originality of this study needs to be clarified.*
*As noted in the specific comments P4 L26 below, the originality of the identification of a mud volcano needs to be clarified. Also, there is an inconsistency of the extent of the study area in Figures 1, 8 and 2, 6, 9. Please confirm that the work outside of the area of interest in Figure 1 is actually carried out by the authors.*

The originality of this study is to highlight the interest of UAS High Resolution DTM to get new active tectonics structures in the Hengchun fault area. We modified locally the text in order to insist on this topic.

We clarified also the way to identifiy a mud volcano from morphostructural analyses.

Dealing with the extension of the study area, we process this ALOS INSAR work during the 2008-2011 period through a common work with Johann Champenois in his PhD work. But at that time numerous political conflicts with the TaiPower Nuclear power plant n°3 prevail and our Taiwanese scientist co-authors asked us not to publish any documents on that area. That is why we limit our study in the PhD to the central and northern part of the Hengchun fault (Fig.8). Since that time, it appears nowadays that it is now possible from a political point of view to publish this major INSAR dataset. That explains the different cover of the figures. Of course, we confirm herein our interest to work in the NHESS paper on the whole onshore Hengchun Fault area that is covered by the UAV survey. For instance, the leveling lines 2 (Fig 9) confirm the LOS INSAR displacement of the Hengchun Fault in the southern part.

*Specific comments:*
*P1 L14: "Characterize" --> "Characterizing"; including this point, the English should be better further corrected throughout the manuscript.*
Done

*P2 L5: "see Fig. 1" --> "Fig. 1"* Done

*P2 L10: Clarify or remove "by previous authors". Also, use "e.g.," instead of "; among others"* Done

*P2 L11: Avoid using "…" (also for the other portions in the manuscript)*
Done

*P2 L20: Explain "PS-InSAR" here. Particularly, the definition of "PS" is missing throughout the manuscript (c.f. P5 L21, 22).* Done

*P2 L23: "few" could be "a few"* Done

*P2 L27: "widly" --> "widely" :* Done

*P3 L4-: Insert a space between the number and unit (e.g., 8cm --> 8 cm).*
Done

*P3 L8: "extracted from airborne LiDAR dataset" Please provide details of this procedure including the location (distribution) and characteristics (what sort of materials) of the ground control points, because this is crucial for the accuracy description. Also, please clarify what does "the open bar ground area" (L10) stand for. Is it a small area for the comparison, or a wide area covering entire dataset? If the former, how is it close to the GCPs?*

We have improved the description and the fig 2, as indicate as follows :

18 ground control points are extracted from airborne LiDAR dataset and from the airborne LiDAR associated 25 cm resolution orthorectified image. Most of the ground

control points situated one the crossroads, targeted and georeferenced from orthorectified image and elevation from the airborne LiDAR data, respectively. The comparison of the UAS DTM with airborne LiDAR data gives a root-mean-square deviation (RMSD) of 4.1 cm with maximum error of 42.5 cm from 26 sites of open bare ground area, e.g. roads, school playgrounds, unvegetated terrains, and parking lots. The elevation of the check point is averaged from an area of 4 m2, equal to the grid size of airborne LiDAR data. The distribution of the ground control points and check points indicate on Fig 2b.

*P3 L27: "e.g.," instead of "among others"* Done

*P4 L5: "work in progress" This wording often appears in this manuscript, but it may not be suitable to regard it as something like a citation. Better to be removed and clarified as a future issue.*
We removed the (only) two "work in progress" written in this paper...

*P4 L26: Please clarify how the mud volcanoes were identified. If they are based on some literature (e.g., Giletycz 2015, NCU dissertation), the original work should be properly taken into account and cited.*
We clarified the way the mud volcanoe were identified from the DTM (Line 29-31 page 4) and we cited Giletycz PhD dissertation 2015. Done

*P5 L26: Be consistent to use the shortened term "HeF": if this is used, it should appear on the former side of this manuscript and thereafter the use of "Hengchun Fault" should be avoided. However, I personally think "HeF" may not be necessarily introduced in the manuscript text (better to be in the Figures only). The term "Hengchun Fault" is not so long.*
As suggested, we use Hengchun Fault in the manuscript (HeF in the 2 figures).

*P6 L1: The methodological description of "GPS measurement" is missing.*
We precise the GPS data used to validate our PS-InSAR results.

*P6 L28: "StaMPS" or "STAMPS"? Be consistent.*
modification done for StaMPS

*Figure 1 caption: "Figure 1a:" --> "(a)", also for b and c.* Done

*Figure 1 GPS station: The displacements of the HENC GPS station indicates an uplift in Figure 1, but why is that in Figure 8 negative? If Z component is positive (uplift), the LOS component should also be positive. Please clarify together with the methodological description of the GPS measurements as suggested above.*

Firstly, we replace fig. 8B by a new figure 8B without the GPS arrows (mistake that we have not seen)...
But within the Hengchun valley there is both subsiding and uplifting places :

1. The Hengchun valley is not homogenous along the Hengchun Fault that lead to relative local subsidence (see north of Hengchun city) and local small uplifts.

2. One may note that PS-InSAR results give a relative displacement contrasting to the absolute displacements given by the GPS measurements. The displacement of GPS HENC stationcorrespond to the absolute displacement which is compared to a PS network referenced toward the chosen PS base (black and white star situated close to Haikou - in the north of the Hengchun Valley). The PS base is chosen with the fewer variability of displacements within the InSAR monitoring time period and appear consequently the more stable area. Anyway this base might be submitted to small continuous uplift or subsidence that may explains local discrepancies with the GPS average annual displacement.

*Figure 1 Area of Interest: The area of interest shown in Figure 1 seems to correspond to that of Figure 8, but in Figures 2 and 6 the study area is much wider including the southern coast. In particular, the mud volcano is located out of the range of the area of interest in Figure 1, and it is unclear whether this was investigated by the authors or derived from something else (see the comments for P4 L26). Moreover, the whole leveling line 2 and the eastern half of line 1 are apparently out of the extent of Figure 8, and it is unclear how the PS values were obtained for these areas. These inconsistencies should be clarified.*

Our study area correspond to the onshore Hengchun Fault covered by the UAS survey acquired and shown in Fig. 2. Some figures present a smaller extension due to potential political conflicts with sensitive Taiwan infrastructures (Nuclear PowerPlant N°3).

*Figure 2: Put (a) and (b) in the panels. Avoid using "Right" or "Left" in the caption.*
Done

*Figure 3: Put (a)-(i) in all the panels. The scale and north direction are missing.* Done

*Figure 4: Put (a)-(c) in the panels. Better to show the photo location and direction in Figure 2.* Done

*Figure 5: The red lines are too thick to prohibit viewing the cracks in the photo. Can they be thinned or set higher transparency?* Done
*Also, "Fang-Shan village" is not shown in Figure 1.*
GPS data was and is still given in the legend.

*Figure 6: It would be better to show the rectangular extents of the example areas of Figure 3 (same as in Figure 1). Red (2) and pink (8) lines are hard to differentiate.*
This important figure 6 is difficult to read and we would like to avoid to

add too many things on it not directly linked to the thematic... that is why we have chosen to draw the quadrangle on Fig. 2.

*Figure 8: If the current A and B show the same displacements, the left one can be omitted. The schematic model of LOS (graphic description including satellite) should be placed in a separate panel, and the flight direction and LOS could be placed in the map panel (like Figure 1).*
Ok we remove the Fig 8A, we redraw that figure

*Figure 9: Put (a)-(c). Explain in caption what the pink area indicates.*

Pink area correspond to the error bar of the leveling data, and denoted on the fig caption, accordingly.

Thanks to take into careful consideration those modifications of our original manuscript.

Chang Kuo-Jen (and Benoit Deffontaines),

---

## Referee Report (RR1)

The reviewer judged that this manuscript is potentially acceptable, but please consider next addition and comments.

Preferable addition

1. Fig. 10b and 10c: line 2 of the caption says "projected on LOS". Please explain in the text how to project three dimensional GPS-measured velocity (dx, dy, dz) into one dimensional LoS velocity (Hanssen 2001), showing equations and values of (dx, dy, dz).

Comment

1. About Figures 11 and 12. The reviewer understood mapping expression of LoS change in Figures 9 and 10; however, the reviewer could not understand how LoS change is able to be projected on vertical (Figure 11) and horizontal (Figure 12). Three-dimensional data are projectable into one-dimensional data, but one-dimensional data such as LoS change does not reveal only vertical and horizontal components of displacement. This paper deals only ascending orbit of PALSAR data, in this case, one-dimensional (elongate or shorten) change is revealed. If the authors calculate not only ascending but descending PALSAR data, they will be able to know the site has uplifting or EW motion (Fujiwara et al. 2000). However, the authors describe "if we generalize, … uplifting toward the LoS" (p.8, L.11) and "uplift" and "subsidence" (p.8, L.16) using Figure 11. Furthermore, the authors describes variations of velocity offset along strike of the fault (p.7, L.26). The reviewer could not understand why such the generalization and assumption is appropriate. Even if photo interpretation support this phenomena, LoS change has information deficiency to support it. The reviewer feels that both photo-interpretation and LoS change is unfairly related and discussed. The reviewer could understand that this paper is not expected to show fault model, but as far as three-dimensional deformation is related to the fault and anticline motion shown in Figure 14, the reviewer thinks that strict evaluation about LoS change is needed. The reviewer guesses that the authors do not have redundant force to calculate descending PALSAR data, or descending PALSAR data may not cover your interesting area. Therefore, how about delete Figures 11 and 12 and re-consider description of assumption and generalization about LoS change in LL.21-28 of p.7, LL.11-19 of p.8, and conclusion?

2. Here, the reviewer does not intend to cast issue in question, but in Figure 8, the authors use PALSAR pair data, whose Bperp is more than 1,500m. In the reviewer's experience, Bperp more than 1,500m gives low coherent result and be barely acceptable. Perhaps the authors consciously eliminate LoS change data derived from Bperp > 1,500m or STaMPS software automatically discard such low-coherent LoS change data, but in the authors' future work, please pay attention to use PALSAR data pairs that have too long Bperp.

References

Fujiwara S, Nishimura T, Murakami M, Nakagawa H, Tobita M, Rosen PA (2000) 2.5-D surface deformation of M6.1 earthquake near Mt Iwate detected by SAR interferometry, Geophys. Res. Lett., 27, 2049-2052.

Hanssen RF (2001) Radar interferometry: data interpretation and error analysis. Kluwer Academic Publishers, Dordrecht

---

## Author Response (AR3)

Please find below our response and modifications that we have revised in the manuscript following the comments and suggestions.

**Editor #1**

**Comment #1-1.** As the Reviewer #1 suggests, the current paper does not fully demonstrate the relationships among the UAS-derived geological interpretation of the fault, independent InSAR LoS interpretation, and the inferred deformation model of the fault. I am not aware of the "classical geological way", but the mixture of methods/results/interpretations for each topic in the current manuscript seems to make it difficult for readers to figure out what is the new point of this study. To avoid this, I would recommend to restructure the entire manuscript (as I originally suggested), or at least, make clear steps in each section of different methods.

**Response #1-1:** Contrasting to what is said, the plan of our paper is as follow: We first began by (1) locate the Hengchun area active faults by doing photo-interpretation of the high resolution UAS DTM composed of alluvial and muddy lithlogies and where no faults are clearly outcropping. Then we focus on (2) the characterization of active faults from the morphostructural analyses checked in the fields, then (3) to quantify the interseismic displacement of those active faults using both leveling, GPS and PSInSAR dataset. We finally insist on the implication of those interseismic active tectonic displacements towards the Taiwan sensitive infrastructures. We believe that our plan is clear, logic and pedagogic.

**Comment #1-2.** Moreover, as listed below, there are too many issues that have not been correctly addressed in the authors' revision. It is necessary to clarify all the issues.

**Response #1-2.** Originally submitted the january 31st 2017, the modification asked by reviewers had been taken into consideration and had been modified through the imposed four major revisions...

**Comment #1-3.** "we confirm herein our interest to work in the NHESS paper on the whole onshore Hengchun Fault area that is covered by the UAV survey. For instance, the leveling lines 2 (Fig 9) confirm the LOS INSAR displacement of the Hengchun Fault in the southern part.

>> This is not true, as the area of interest of InSAR shown in Fig. 1 is apparently different from the area of leveling survey shown in Fig. 9. Eastern half of both of the leveling survey lines are out of the range, but still values of LoS are given along the line (Fig. 9b, c).

**Response #1-3:** We follow your suggestions and modified the figures in order to get the same studied area for all the paper. Consequently this comment seems out of scope with this new version.

**Comment of the Response #1-3**: Fig. 10 covers wider area than the areas of interest shown in Figs. 1, 2, 7, and 9. The eastern half in Fig. 10 could therefore be eliminated.

**Response of the new comment #1-3:**

**-** The reason that we have adopted the leveling technique is that it shows good resolution of the vertical component. To verify the InSAR data, we need to have a level reference survey. It is therefore suggested not to eliminate the data and figure of eastern Hengchun Peninsula, as the easternmost and westernmost reference points are on an enclosed curve line of the Class 1 level reference net defined by the country's Satellite Survey Center of Department of Land Administration. For detailed information, please refer to the following website: http://gps.moi.gov.tw/SSCenter/Introduce_E/IntroducePage_E.aspx?Page=Height_E6
- In addition, Fig. 10 gives the two leveling lines existing across the Hengchun Peninsula. It is important to publish the entire section as it reveals the place where are located the active faults in the Hengchun Peninsula associated with vertical components. especially for the Kenting Fault situated East of the Hengchun Fault. That is why we do insist to publish the Fig10 in its original geometry. Anyway, we redraw fig.10 in a better definition and we precise the location of both Hengchun and Kenting Faults.

**Comment #1-4** P2 L10: Clarify or remove "by previous authors". Also, use "e.g.," instead of "; among others" and When "e.g.," is used, "etc." is unnecessary.
**Response #1-4:** Done

**Comment #1-5**. P2 L11: Avoid using "…" (also for the other portions in the manuscript)
Done >> Not corrected. It remains in many places throughout the manuscript.
**Response #1-5:** Corrected.

**Comment #1-6** P2 L20: Explain "PS-InSAR" here. Particularly, the definition of "PS" is missing throughout the manuscript (c.f. P5 L21, 22).
Done
>> Still missing is the explanation of InSAR (at least, it should be spelled out).
**Response #1-6:** Now, it is spelled.

**Comment #1-7** P4 L5: "work in progress" This wording often appears in this manuscript, but it may not be suitable to regard it as something like a citation. Better to be removed and clarified as a future issue.

We removed the (only) two "work in progress" written in this paper >> >> Not corrected. There still remains this wording in the manuscript.

**Response #1-7:** We had removed all 'work in progress' in the Manuscript. Moreover, we remove anything which could give working perspectives in this area.

**Comment #1-8** P6 L1: The methodological description of "GPS measurement" is missing.

We precise the GPS data used to validate our PS-InSAR results.

>> The details of the "GPS measurement" is still unclear. The authors seemed to use data from static GNSS stations for revealing displacements, as well as kinematic GNSS measurements for GCPs, but these details are not properly provided (some missing and some unclear). Also, detailed description of airborne LiDAR is missing. As noted above, clear descriptions of the methodologies, as well as their results and interpretations, are necessary to be provided separately.

**Response #1-8:** A paragraph had been added with a new figure (Fig.3), As this paper is not focus on GPS paper, we did not emphasize too much on it, the reference Yu S.B., et al. 1997 is here to fulfill further queries.

**Comment of the Response #n1-8:** Response #1-8: Even if the focus of this paper is not on GPS (I would recommend this to rephrase as GNSS, a more generic term, if some other satellite data are used such as GLONASS) itself, the only one reference is insufficient to correctly represent the methodology. Not necessarily to be too detailed, but the authors can provide some more details for an objective validation.

**Response of the new comment #n1-8:** Contrasting to the reviewer 1, as we used the Yu S.B. et. al., 1997 results deduced from its dataset and methodology, so (1) we confirm that GPS is herein the right term; (2) we invite NHESS readers to look at Yu et al. 1997 Tectonophysics paper and the work of his team to get further GPS informations on the used methodology. Anyway we insert in the txt some of the Yu S.B. methodology.

**Comment #1-9** Figure 1 caption: "Figure 1a:" --> "(a)", also for b and c.

Done >> Not done.

**Response #1-9:** We redraw the Fig 1 and with a, b, c…

**Comment #1-10** Figure 1 Area of Interest: ... Moreover, the whole leveling line 2 and the eastern half of line 1 are apparently out of the extent of Figure 8, and it is unclear how the PS values were obtained for these areas. These inconsistencies should be clarified.

**Response #1-10:** We modified the study area in order to have the same for all figures. All the datasets had been processed in the same time.

**Comment #1-11** Our study area correspond to the onshore Hengchun Fault surroundings partly covered by the UAS survey acquired and shown in Fig. 1 and 2. Some figures present a smaller extension due to potential political conflicts with sensitive Taiwan infrastructures (Nuclear Power Plant N°3). >> This is not the point, as noted above at (1)

**Response #1-11:** We fully disagree with the answer given herin "this is not the point" as It was due to political reasons some of our co-authors did not want to publish those results highly sensitive. Anyway, due to the modifications of the extension of the figures this comment is now out of scope as fig.3a and 10 cover effectively the southern Hengchun Peninsula leveling part as well as the Hengchun/Kenting Faults surroundings.

**Comment #1-12** Figure 4: Put (a)-(c) in the panels. Better to show the photo location and direction in Figure 2.

Done >> Not corrected. Also, Figure 2 shows the photo location but the direction is unclear.

**Response #1-12:** The Fig.6 (photograph in the fields) was out of the perimeter of former Fig.1 that is why we added in the previous version submitted in july 2017 the GPS coordonates. As the readers may want to see the exact location of Fig.6, we enlarge to the north the Fig1 in order to locate the outcropping fault (red dot) and the Fang-Shan Village.

**Comment #1-13** Figure 5: … Also, "Fang-Shan village" is not shown in Figure 1.

GPS data was and is still given in the legend. >> >> It is never seen anywhere else in the manuscript.

**Response #1-13:** Fang-Shan village is added on Fig.1 in this new version.

**Comment of the Response #n1-12, 13**: Response #1-12,13: Fang-Shang (Figure 6 caption) or Feng-Shang (Figure 1 caption)?

**Response of the new comment #n1-12, 13:** We used herein on this version only "Fang-Shan" city.

**Comment #1-14** Figure 6: It would be better to show the rectangular extents of the example areas of Figure 3 (same as in Figure 1). Red (2) and pink (8) lines are hard to differentiate.

This important figure 6 is difficult to read and we would like to avoid to add too many things on it not directly linked to the thematic... that is why we have chosen to draw the quadrangle on Fig. 2. >> Still I cannot clearly identify the pink (8) lines.

**Response #1-14:** We removed Pink 8 lines in the legend as they were difficult to see on the W coast... see original figure of first submission 31st january 2017.

**Comment #1-15** I understand putting less information is better, but still do not understand the correspondence between the coverage by the UAS-derived data and that by the "morphostructural map". Does the western margin of "9: Hengchun valley alluvial and marine deposits" overlaps with the area? If not, how was it mapped?

**Response #1-15:** the western part of the index 9 has been mapped from the concave shape of the 5m DTM which is situated in transparency in the background of this figure 9. This drawing is basic and common sense from geological mapping: mapping the external limit of flat soft lowlands that correspond to alluvial and marine deposits in such environments. Few of the region outside of the UAS mission area, the existed low resolution DTM was inferred, e.g. from the 5m DTM.

**Comment #1-16** Figure 8: If the current A and B show the same displacements, the left one can be omitted. The schematic model of LOS (graphic description including satellite) should be placed in a separate panel, and the flight direction and LOS could be placed in the map panel (like Figure 1).

Ok we remove the Fig 8A, we redraw that figure >> Panel A was removed but the other points were not addressed.

**Response #1-16:** We separate the different drawings with specific independent quadrangle. Thanks to confirm it has been done in this new version.

**Comment of the Response #n1-16**: Please provide the satellite orbit path and the line of sight directions projected on the map.

**Response of the new comment #n1-16:** One may note that the original submission version (january 31st 2017) had already both ascending satellite orbit path with the right lateral LOS directions projected on the Hengchun Peninsula (see top right quadrant). Anyway we added values in the manuscript.

**Anonymous Referee #3**

**Referee #3-1.** The main two resulting products are newly interpreted Hengchun Fault in Fig. 6 and the offset of mean LoS velocities between eastern and western part from

the Fault. And the authors make an assertion of the active inter-seismic tectonic deformation model of the Fault.

**Response #3-1:** The purpose of this NHESS submitted paper is to locate, characterize and quantify for the first time the active Hengchun Fault in Southern Taiwan by combining GPS, levelings, PSInSAR and by a detailed photo-interpretation of the high resolution and high precision UAS-DTM and orthophoto of the Hengchun fault area through a precise Morphoneotectonic map of the Hengchun Fault. Contrasting to what was known and published before (see the references), we reveal and quantify herein for the first time the active displacement that affect the Taiwan Nuclear Power plant N°3 and its surroundings.

**Referee #3-2.** However, the newly interpreted Fault plays a minor role in the proposed tectonic deformation model. Even if detailed distribution of the Fault is revealed using UAS products, the authors do not explain how the distribution effectively works in order to propose the tectonic deformation model nor mention how the distribution differentiates the new model from the previous models. Relation between the detailed interpretation of the Fault and the proposed model is vague**.**

**Response #3-2:** To this point, the authors listed 3 points as follows:
1. The aim of this paper is not a structural and tectonic analysis in, and only in, the fields of the Hengchun fault as it is not possible to figure out the full frame of the study area from outcropping. Classical microtectonic studies are not possible to carry on in the muddy Kenting Melange and along the alluvial deposits of the Hengchun fault. That is why we develop in this NHESS paper a new approach based on a combination of morphotectonic approaches that associate to various complementary qualitative and quantitative observations.
2. This paper focus on the new inputs of UAS and its derivative products (DTM and orthomosaic) and their structural and tectonic interpretations (morphoneotectonic map) combined to GPS, PSInSAR interferometry and leveling to better understand the active Hengchun Fault in Southern Taiwan. One may note that this paper was not associate with an EGU tectonic session ! If the paper was dedicated to a tectonic session it would have been written differently with different dataset...
3. On the other hand, the global and continuous uplifting through time and subsiding on both sides of the Hengchun fault deformation of Fig. 10a, b, c and Fig.11 and 12 reveal the progressive folding and the coherence of the proposed tectonic model in this NHESS issue.
4. Contrasting to what is said in #3.2 we newly locate, characterize and quantify the

2008-2011 interseismic activity of the Hengchun fault. In addition, we propose a coherent and common sense model that fits with our dataset and previous works.

**Referee #3-3.** The authors made a crucial mistake that LoS change stands only for vertical component of deformation, i.e., uplift and depression. LoS changes include not only vertical component but also horizontal component of the deformation, however, the authors do not explain enough why vertical component should be paid attention and why horizontal component is not considered.

**Response #3-3:** Indeed, the InSAR result reveals only the 1D LoS deformation. In order to confirm the displacement revealed by PS-InSAR in the Southern Hengchun Peninsula, we integrated other geodetic techniques including: fixed GPS stations, levelings, InSAR and UAS data, so as able to decipher the activity deformation of the Hengchun fault and nearby area. The PSInSAR results are compared to 3 fixed absolute GPS data (Fig.9, HENC, GS57 and GS59) and two E-W leveling profiles (Fig.10a, b, c) that compare LOS and vertical displacements. Our result is fully coherent and evidenced clearly a simple deformation of the Hengchun Peninsula (see new Fig. 7, 10a, b, c, 11, 12, and our active tectonic model - Fig.14). The GPS and the two leveling profiles of Fig.10a, b, c, and 11 reveal the vertical component of the active inter-seismic Hengchun fault displacement and its surroundings during the same InSAR monitoring time period. Please read carefully the manuscript. Thus, there is no "CRUCIAL MISTAKE", we are fully able to differentiate the planimetric and vertical absolute component through GPS fixed stations, leveling and combined to the LOS PSInSAR data displacement.

**Referee #3-4** Fig. 4 infers left lateral movement and Fig. 5 infers EW compression, however, the reviewer could not catch the relation between the newly interpreted Fault and such the field observations. Furthermore, the authors did not mention how the horizontal component of the deformation is interpreted from the LoS change velocity to propose the new model.

**Response #3-4:** Transpressive motion is the structural/tectonic term where both thrusting and lateral strike-slip motion prevail on the same tectonic fault. It is a basic structural and tectonic notion associated to partitioning of the deformation used in this paper to explain the displacement of the Hengchun Fault which is both transpressive and left-lateral.
Contrasting to what is said The Chelungpu fault that was reactivated during the Chichi earthquake present both this left-lateral transpressive displacements (see the

references on chichi earthquakes). Effectively the motion of the Hengchun Fault of Fig.5, 9, 14 is deduced from the GPS fixed station represented in Fig.1. It is common sense and we cited the paper of Chang et al (2003). In our NHESS paper, the authors illustrated and documented carefully Fig.6: the lateral component in between the hanging wall and the footwall wall of the outcropping fault confirming both a compressive motion and a lateral motion (see Fig.6) that we propose in the regional geodynamic model of Fig.14. And confirmed by all field geodetic measurements (GPS, Levelings, PSInSAR and Field Work).

Contrasting to what is said by **reviewer #3-4**: the Fig. 4 shows on Hengchun Fault both compressive illustrated by the first line of the figure (see a, b,c,d), and left-lateral motions (see 2nd line of the figure: e,f,g,h). Fig. 4 and Fig. 5 are coherent and common sense toward GPS datasets (Chang et al. 2003, and this work).

**Referee #3-5.** Therefore, the reviewer thinks that it is difficult for the reviewer to judge the acceptance.

**Response #3-5:** To summarize, in this NHESS paper our PSInSAR results are compared to GPS absolute deformations (Fig.1, 10a,b,c and 11), and two field levelings (Fig. 10a,b,c and 11) that give with no doubt the vertical component of the deformation of the Hengchun fault and of the whole southern peninsula. All is coherent and reveal the simple tectonic model of Fig.14. Of course the deformation deduced from the PSInSAR is only 1D and along the Line of Sight (LoS). That is the reason why in this study, we integrated many other aspects including: GPS, leveling, InSAR and UAS data, so as to be able to decipher the active deformation of the Hengchun Fault and nearby area.

**New comments:**

The reviewer judged that this manuscript is potentially acceptable, but please consider next addition and comments.

**Referee #n3-6.** (Preferable addition): Fig. 10b and 10c: line 2 of the caption says "projected on LOS". Please explain in the text how to project three dimensional GPS-measured velocity (dx, dy, dz) into one dimensional LoS velocity (Hanssen 2001), showing equations and values of (dx, dy, dz).

**Response #n3-6:** In order to precise our methdology, we added the following sentence: "For each of the 3 fixed GPS stations (HENC, GS57, GS59), it has been calculated an average displacement projected into the radar LOS by taking into consideration the various local incidence angle along the distance axis (Hanssen, 2001)." the reference had been added in the bibliography.

**Referee #n3-7**. About Figures 11 and 12. The reviewer understood mapping expression of LoS change in Figures 9 and 10; however, the reviewer could not understand how LoS change is able to be projected on vertical (Figure 11) and horizontal (Figure 12). Three-dimensional data are projectable into one-dimensional data, but one-dimensional data such as LoS change does not reveal only vertical and horizontal components of displacement. This paper deals only ascending orbit of PALSAR data, in this case, one-dimensional (elongate or shorten) change is revealed. If the authors calculate not only ascending but descending PALSAR data, they will be able to know the site has uplifting or EW motion (Fujiwara et al. 2000). However, the authors describe "if we generalize, … uplifting toward the LoS" (p.8, L.11) and "uplift" and "subsidence" (p.8, L.16) using Figure 11. Furthermore, the authors describes variations of velocity offset along strike of the fault (p.7, L.26). The reviewer could not understand why such the generalization and assumption is appropriate. Even if photo interpretation support this phenomena, LoS change has information deficiency to support it. The reviewer feels that both photo-interpretation and LoS change is unfairly related and discussed. The reviewer could understand that this paper is not expected to show fault model, but as far as three-dimensional deformation is related to the fault and anticline motion shown in Figure 14, the reviewer thinks that strict evaluation about LoS change is needed. The reviewer guesses that the authors do not have redundant force to calculate descending PALSAR data, or descending PALSAR data may not cover your interesting area. Therefore, how about delete Figures 11 and 12 and re-consider description of assumption and generalization about LoS change in LL.21-28 of p.7, LL.11-19 of p.8, and conclusion?

**Response #n3-7:** We have divided the above section into 7 separate questions (a to g), and responded these questions accordingly.

**a)** About Figures 11 and 12. The reviewer understood mapping expression of LoS change in Figures 9 and 10; however, the reviewer could not understand how LoS change is able to be projected on vertical (Figure 11) and horizontal (Figure 12).

**Response #n3-7a:** We did not project the 1D PS-InSAR LOS into vertical and horizontal... Since the first submission all our PS-InSAR result is within LOS. Just we used Levelings that give the vertical component + the result of 3 fixed GPS stations situated on both sides of the Hengchun fault which gives by analogy the same profile geometry (see Fig. 10, and 11). That is why those 2 figures are of key interest and cannot be removed.

**b)** Three-dimensional data are projectable into one-dimensional data, but one-dimensional data such as LoS change does not reveal only vertical and horizontal components of displacement....

**Response #n3-7b:** This is true if you have only PS-InSAR dataset which is not the case herein (South Hengchun Peninsula) as we also have got levelings and GPS fixed stations that give us all the component of the deformation during the monitoring time period.

**c)** This paper deals only ascending orbit of PALSAR data, in this case, one-dimensional (elongate or shorten) change is revealed. If the authors calculate not only ascending but descending PALSAR data, they will be able to know the site has uplifting or EW motion (Fujiwara et al. 2000). However, the authors describe "if we generalize, … uplifting toward the LoS" (p.8, L.11) and "uplift" and "subsidence" (p.8, L.16) using Figure 11. Furthermore, the authors describes variations of velocity offset along strike of the fault (p.7, L.26). The reviewer could not understand why such the generalization and assumption is appropriate.

**Response #n3-7c:** As mentionned herein and unfortunately, we do not have the ALOS descending orbit on the Hengchun Peninsula, so we cannot use it to better constrain the radar displacements. But we note that the resulting displacement of the 3 fixed GPS stations situated on both sides of the Hengchun Fault projected in the local radar LOS, and the two vertical Levelings (L1 and L2) acquired in the same time period are coherent with the LOS PS-InSAR displacements. We consequently conclude that the general horizontal displacement is not that important and the LOS displacement is close to the vertical one.

**d)** Even if photo interpretation support this phenomena, LoS change has information deficiency to support it...

**Response #n3-7d:** Effectively the PS-InSAR LOS itself and alone is not able to support it but we have other geodetic information that we have taken into account in our interpretation...

**e)** The reviewer feels that both photo-interpretation and LoS change is unfairly related and discussed....

**Response #n3-7e:** We publish the new morpho-structural interpretation of the Hengchun Peninsula (which was not done before) with the help of this UAS High resolution survey. Consequently, we propose characterization and quantification for the Hengchun Fault with a simple model that fits with the available dataset. So we disagree with this remark of the reviewer but we agree that more work needs to be

done to clarify the displacements of the Kenting Fault and the eastern part of the Hengchun Peninsula. We wanted to write it in the Ms but each time it was asked to remove it...

**f)** The reviewer could understand that this paper is not expected to show fault model, but as far as three-dimensional deformation is related to the fault and anticline motion shown in Figure 14, the reviewer thinks that strict evaluation about LoS change is needed. The reviewer guesses that the authors do not have redundant force to calculate descending PALSAR data, or descending PALSAR data may not cover your interesting area....

**Response #n3-7f:** Yes you are right, it is the case.

**g)** Therefore, how about delete Figures 11 and 12 and re-consider description of assumption and generalization about LoS change in LL.21-28 of p.7, LL.11-19 of p.8, and conclusion?

**Response #n3-7g:** Of course not. We will not delete those figures that are one key of this paper for the reasons explained above: we have got not only PS-InSAR data but also levelings and GPS dataset that help to constrain the PS-InSAR dataset and help us by comparison with the field structures to explain the deformation of the Hengchun Peninsula.

**Referee #n3-8.** Here, the reviewer does not intend to cast issue in question, but in Figure 8, the authors use PALSAR pair data, whose Bperp is more than 1,500m. In the reviewer's experience, Bperp more than 1,500m gives low coherent result and be barely acceptable. Perhaps the authors consciously eliminate LoS change data derived from Bperp > 1,500m or STaMPS software automatically discard such low-coherent LoS change data, but in the authors' future work, please pay attention to use PALSAR data pairs that have too long Bperp.

**Response #n3-8:** Of course, if you process DInSAR radar images with a Bperp > 1km are barely to used, but the interest of PS-InSAR processing (Stamps) allows one to extend a bit the values of perpendicular baselines to a max of 2.5km- see Hooper A., 2009).

**Anonymous Referee #4**

**Referee #4-1.** Somewhere in discussion could you emphasize the improvement of using the UAS derived DTM than the 40m DEM? I cannot easily see what can only be resolved with using the new DTM.

**Response #4-1:** We add the Fig.3 and a paragraph on it, explaining the differences.

**Referee #4-2.** Page 2, Line 10: I am not sure whether using exclamation point is the best choice of punctuation in this sentence.
**Response #4-2:** We remove it.

**Referee #4-3**. Page 2, Line 13: Indicate the power plant in Fig. 1.
**Response #4-3:** We add it, See orange circle (3)

**Referee #4-4** Page 2, Line 21: Write down the full name of InSAR when first mentioned it.
**Response #4-4:** we add: Persistent Scatterers-Interferometry Synthetic Aperture Radar (PS-InSAR hereafter)

**Referee #4-5**. Page 3, Line 8: Delete "…".
**Response #4-5:** We removed in the text of 3 places where "…" remained

**Referee #4-6**. Page 3, Line 9: "Eighteen" instead of "18" in the beginning of a sentence.
**Response #4-6:** now it is: Eighteen (18) ground control points

**Referee #4-7**. Page 5, Line 12: "in" Figure 6
**Response #4-7:** Done, and now is in Fig. 7.

**Referee #4-8**. Page 5, Line 21: Need a full citation of Chen's report.
**Response #4-8:** We cited it and add it in the reference.

**Referee #4-9**. Section 4: Where is the reference point for the PS analysis? Please indicate the location in Fig. 8.
**Response #4-9:** Reference points for the PS analysis are white star (4) in Fig 9, it is (and it was) indicated.

**Referee #4-10**. Page 6, Line 1: might be submitted to → might be subject to.
**Response #4-10:** Done

**Referee #4-11**. Page 6, Line 16: Dealing with the Kenting → "Regarding the Kenting" or "With regard to Kenting".
**Response #4-11:** Regarding the Kenting Fault…

**Referee #4-12**. Page 6, Line 25-26: Do you mean that the GPS LOS velocity is converted from leveling measurements and GPS horizontal measurements?.

**Response #4-12:** No, GPS measurements are deduced directly from fixed stations existing in Taiwan. Levelings is a completely independant work and dataset had been processed by Lin Kuan-Chan and Hu Jyr-Ching (co-authors of this NHESS paper).

**Referee #4-13**. Page 6, Lines 14 and 17: Please avoid using "…".

**Response #4-13:** Done

**Referee #4-14**. Page 7, Lines 26: How is the creeping value estimated? If you refer to the difference of surface velocity between the west and east of the Hengchun fault (in stead of creeping along the fault), I will suggest saying "difference in interseismic velocity between west and east of the Hengchun fault with a value of 8 mm/yr". Based on the figure it looks like 8 mm/yr instead of 0.8 mm/yr..

**Response #4-14:** We agree, we modify it as the reviewer proposed. Yes we put all values of velocities in mm/yr (avoiding cm/yr). We change all velocity units in mm/yr. BUT DTM Resolution and precision are still in cm...

**Referee #4-15**. Page 8, Lines 6-8: I know this is a schematic model figure, but could you infer the dip angle along strike of the Hengchun fault? Is it a high or low angle fault?.

**Response #4-15:** Please refer to the cross-section in Fig.1c, and the Hengchun is vertical due to the rectilinearity mapping, Kenting Fault is low dipping fault so as to demonstrates its sinuosity. It was (and is still) also explained in the text, page 8 Line 22. We add Hef... "an almost vertical" fault.

**Referee #4-16**. Page 8, Line 12: present both left-lateral strike-slip and thrust dip-slip components such as ....

**Response #4-16:** Done.

**Referee #4-17**. Page 8, Line 13: Could you provide more examples of faults with this oblique component property?

**Response #4-17:** We modified as: Chelungpu Fault, etc. (Deffontaines et al. 1997)

**Referee #4-18**. Page 8, Line 14: Please avoid using both "cm" and "mm" in the same paper.

**Response #4-18:** Done. We change all velocity units in mm/yr. BUT UAS DTM

Resolution and precision are still in cm.

**Referee #4-19.** Page 8, Line 116-18: I think this comparison is a bit unfair, as the InSAR derived velocity is in LOS, whereas vertical from marine terrace dating results. If the authors know the fault dip angle they should try fault inversions using different datasets, and then compare the inferred slip in geologic and geodetic time scales. Also I suggest saying "geodetic slip rates" instead of "instantaneous slip rates

**Response #4-19:** As levelings give us the vertical component of the deformation, the GPS fixed station give us the absolute deformation, thus it is possible to compare the marine terrace dating results. And consequently it is possible to make this comparision.

Yes it is possible to inverse the deformation dataset as we know the Fault dips. However, it is not the aim and the scope of this UAS/NHESS paper. By the way, we are doing this tectonic work independently on a global study of the whole Hengchun peninsula, and planning to submit the study in a Structural/Tectonics journal.

"geodetic slip rates" modification done.

**Referee #4-20.** Page 9, Line 14: PS km-2.
**Response #4-20:** Done.

**Referee #4-21.** Page 9, Line 16: the highly dipping Hengchun Fault → the Hengchun Fault with high dip angle.
**Response #4-21:** "the Hengchun Fault with high dip angle" Done.

**Referee #4-22.** Page 9, Line 16: Choose either "interseismic" or "inter-seismic" throughout the manuscript.
**Response #4-22:** Done, "interseismic" is used in this manuscript.

**Referee #4-23.** Page 9, Line 23: due to (1) xxx, (2) xxx..
**Response #4-23:** Done.

**Referee #4-24.** Page 9, Line 23: to the low fault dip angle deduced from ….
**Response #4-24:** Done.

**Referee #4-25.** Page 9, Line 27: Suggest using "Nevertheless" than "Anyway".
**Response #4-25:** Done. "Anyway" replace by "Nevertheless".

**Referee #4-26.** Figure 3, Line 9: I cannot find NPP and MV in the figure.

**Response #4-26:** We indicate NPP on Fig.1, but removed MV.

**Referee #4-27.** Figure 8, Looks like the PS points in the southernmost part are not plotted, as they are shown in Fig. 9 in comparison with leveling measurements.
**Response #4-27:** We modified and redraw Nearly all figures in order to have the same studied area... Please see Figs. 1, 2, 3, 4, 5, 9, 10, 12, 14.

**Referee #4-28.** Figure 9: How the InSAR error bars were estimated?
**Response #4-28:** 90% of confidence.

**Referee #4-29.** Figure 10, Is the label (2) needed in this figure?
**Response #4-29:** Effectively It has been removed.

**Referee #4-30.** Figure 11, Is this figure showing offset between the hanging wall and the footwall? Where is the GPS measurement (HENC?) relative to?
**Response #4-30:** Yes. The GPS measurements (HENC) is relative to a Penghu-Taipei Line (see Yu S.B. et al., 1997). However, we do not develop too much this aspect as it is not the topic of this paper.

**New comments:**
**Referee #n4-31.** The quality of the revised manuscript has been improved, but I think the authors should consider rewriting their discussion and conclusions. Conclusions should be more or less a summary of their work without "new" statements or discussion, but I find a large portion in their conclusion section is in fact discussion.
**Response #n4-31:** We modified the end of our ms by adding only one paragraph on "discussion and conclusions

**Referee #n4-32.** In section 4, I think the authors need to add more details about their PSInSAR processing. It is not clear to me how the super master is determined. How they set up the coherence threshold, what kind/resolution of DEM they used (SRTM DEM?), how they determine the reference point (a single point or average of multiple points close to the station GS59?)

**Response #n4-32:** We have divided the above section into 4 separate questions (a to d), and responded these questions accordingly.

a) think the authors need to add more details about their PSInSAR processing. It is not clear to me how the super master is determined.

**Response #n4-32a:** The super master is determined by the mid-time serie as it was said in the text.

**b)** How they set up the coherence threshold
**Response #n4-32b:** Stamps specific processing.

**c)** What kind/resolution of DEM they used (SRTM DEM?)
**Response #n4-32c:** No, we used a local 40m ground resolution DTM.

**d)** How they determine the reference point (a single point or average of multiple points close to the station GS59?)
**Response #n4-32d:** Only one point but characterized by a continuous no to small displacement during the monitoring time period. etc.

**All Response #n4-32:** We did not intend to highlight too much on that technical points because of the points are well documented in Champenois PhD Thesis (2011A), and it is not the scope of this NHESS special UAS issue. We added references in the text to fill in that point.

**Referee #n4-33.** One other concern I have is that there is no clear connection between their new DTM and the results from PSInSAR. To me, they are two separate studies in a paper. Additionally, it is not clear to me what can only be resolved with the new 0.13m DTM if not already done with the 3m DEM. It seems to me the geologic interpretation in Figs 4 and 14 can already be drawn with the 3m DEM.
**Response #n4-33:** Effectively we did not use the HR DTM to make the PS-InSAR processing as it does not cover the same area. Since last january we are enlarging the acquisition area but it takes time. Of course as we have the whole Kenting Fault area and W and E Hengchun Peninsula, we will update the processing with the corresponding radar images and geodetic data...
Fig. 3 should answer this question as Fig.3 shows clearly the great technical differences in the resolution and the precision of the different DTM's. Fig.4 show definitely the input of a High resolution DTM even toward a 5m /3m or even to a 1m ground resolution DTM.

**Referee #n4-34.** This manuscript can be more interesting if the authors can demonstrate improvement of the understanding of the Hengchun and Kenting fault system by using both their high resolution DTM and PSInSAR. Something like identifying creeping/locked portion of the fault, surface evidence of the fault scarp

from high resolution DTM, or inferred fault locking depth from using both DTM and PSInSAR will significantly make the study more interesting. I think it is also relevant to the scope of this journal.

**Response #n4-34:** Thanks for this remark as it is the aim and the scope of our paper. We locate through morphostructural interpretation the location of the active faults. Then we characterize them using geodetic and PS-InSAR datasets and we follow alongstrike the Hengchun Fault and evidence places where there is slow and active creeping areas. There is still work to be done for the Kenting Fault more complex to decipher.

**Referee #n4-35.** Page 6, lines 17-18: This sentence doesn't read well.

**Response #n4-35:** We modified "... The PS-InSAR base (black and white star) is chosen with the more stable place in the figure (very few displacements - see close to Haikou - north of the Hengchun Valley) within the monitoring time period."... as follow :

The PS-InSAR base (fixed GPS station: GS59, correspond to the black and white star see its location close to Checheng - N of the Hengchun valley, on Fig.1 and 9) is chosen as it presents a stable to very low continuous deformation during the monitoring time period...

**Referee #n4-36.** Page 10, lines 5-6: It could be due to folding or the shallow part of the fault is locked, right?

**Response #n4-36:** Not as clear... In the northern part the Hengchun Fault is narrow and clearly outcrops beolw the marine terraces with a measurable offset (see Fig. 4d). That contrast with the central and southern part where the Hengchun Fault is wider and submitted to active folding and only locally the fault is partially locked.

**Referee #n4-37.** Figure 9: Please increase the size of the PS points and/or adjust color scale. It is hard to see the PS points.

**Response #n4-37:** It is the maximum we can do to adjust the color for the PS points. There is no transparency. We tried to enlarge those but then they superimposed a lot and the result is confusing. In addition the DTM show an interesting contrast to help location. This was our choice.

**Referee #n4-38.** Page 7, lines 21-23 and Figure 12: As mentioned before, I think a better plot is taking the differences between hanging wall and footwall of the Henchung Fault. In this way it can better characterize the along-strike fault creeping behavior.

**Response #n4-38:** That is what we have done. We have taken the difference of altitude of the mobile average of both points situated on both side of the fault zone (see Champenois et al., 2012).

**Referee #n4-39**, Page 9, line 26: submitted also → also subject
**Response #n4-39: done**

Finally, we added references in the bibliography and we modified the list of authors, (removal of Benedicte Fruneau from the list of authors). Acknowledgements with Erwan Pathier and Benedicte Fruneau.

---

## Author Response (AR4)

We thank the Editor for the comments that allowed us to clarify and improve the contents of the manuscript. Our answers to the Editor's comments are listed by a point-by-point basis as follows. The blue text in the manuscript indicates the revisions that we made in previous comments. The red text in the manuscript indicates the changes that we further made after the latest comments by the Editor.

**RE #1-3, #1-10, and #1-11:** There may be some misunderstandings. The problem has been the inconsistency among the coverage areas for the data provided by InSAR and leveling surveys. If the InSAR data is available for the eastern area as the authors suggest in their response (this was not clear in the manuscript), Figure 1 should show the entire extent of the area of target, including the eastern side (like the case for the current Figure 3a). It is also necessary to explicitly indicate the extent of available data for InSAR, like that for the UAS data area. Figure 9 should also be widened to include the eastern side, together with the PS-InSAR plots and leveling lines. If this is achieved, current Figure 10a becomes unnecessary (i.e., all the locational contents in Figure 10a can be integrated into Figure 9).

**Answer:** As we focus herein on the Hengchun Fault (which is situated W of the Hengchun peninsula), we provided in this paper only the PS-inSAR dataset on the studied area. Finally, we modified the leveling lines as the reviewer insist and we modified the eastern part of Fig.10 !

**RE #n1-8:** I do agree with the authors that too many details are unnecessary for the cited work. However, one may not be able to instantly read the cited paper, and it should be helpful and more scientifically sound if some basic descriptions of the method (including measurement type (static I guess), accuracies, temporal frequencies, measurement periods, etc.) are provided.

**Answer:** Effectively, we focus in this paper on the location, characterization and quantification of the Hengchun active fault. We unfortunately are not writing a paper on leveling, GPS or PS-inSAR, so we only used those dataset and cited the good authors and the NHESS reader is deeply invited to go to those references to get further information not dealing wand away of the active tectonic thematic.

**RE#n4-33:** Please revise the newly added Figure 3 with regard to the better interpretation of the geological components (geological boundaries, fault lines, and terraces). It is still unclear how these high-resolution images contribute to the better interpretations of the geology in this area, although this is partly shown in the next Figure 4. Therefore, please provide some interpreted lines on these high-resolution maps in Figure 3.

**Answer:** Despite we disagree with this reviewer comment, as Fig 3 was dedicated to the comparison of the numerical topography dataset quality. That is why we compared different topographic DTM resolution (40m up to our new 7cm ground resolution). We added the few fault lines that was asked. In contrast Fig 4 is dedicated to the morphostructural interpretation with raw data and geological mapping in 2D and a schematic 3D view. The Fig. 4 is definitely there to highlights some of the pedagogic morphostructural interpretation. One may note that any reviewer is invited to read the basic of morphostructural interpretation from topography if he wants to learn more about how to get structures from topography.

**Some other editorial comments:**

**NE#1**. Abstract: It is better to remove citations in Abstract (P. 1, L. 19).
**Response:** We removed CPC and CGS...

**NE#2.** P. 1, L. 23 and 33: Please spell out GPS.
**Response:** We added Global Positioning System (GPS)

**NE#3.** P. 2, L. 1: Please provide the unit in mm/y.
**Response:** We modified the GPS displacement in mm.y$^{-1}$ as required.

**NE#4**. P. 2, L. 19 and followings, on the terminology for DTM, DSM, and DEM: There seem mixed uses of these three words. In general, DEM (digital elevation model) is a generic term, including the filtered DTM (digital terrain model) showing bare land and surficial DSM (digital surface model) before filtering/removing ground objects. Photogrammetric approaches can usually provide only DSM (except some cases), while airborne lidar can provide both of DTM and DSM. In particular, UAS usually generates DSM only. If DTM is generated from UAS-derived photogrammetric data, there should be some specific algorithms to filter the raw point cloud. Anyway, please provide the definition of these terms when first appears in the text, and consistently use the appropriate ones throughout the manuscript.
**Response:** We have now defined the terms in P. 4, in this manuscript as suggested. (Note: The Taiwan government and the geomatics community define the term DTM as a general term, whereas the DEM was defined as the geomorphologic elevation after removing the buildings, trees and vehicles, etc. DSM defines the first return of the LiDAR pulse of terrain data, including buildings and tree canopy). The UAS generates DSM usually, for some case DTM can be produced from dense point cloud. The wordings in the manuscript are clearly defined now.

**NE#5**. P. 2, L. 26: "PI JAXA…" This information should better be provided in Acknowledgements with more details, not in the main text.

**Response:** This information is removed page 2 as it was already in the acknowledgements

**NE#6**. P. 2, L. 29: Please provide the webpage with URL in a citation format.

**Response:** URL is removed in the text and modified by "Academia Sinica GPS..." and added in the reference list.

**NE#7**. P. 3, L. 16: Remove "(18)" after "Eighteen".

**Response:** 18 is removed

**NE#8**. P. 3, L. 16-17: Please spell out "LiDAR".

**Response:** Light Detection And Ranging added

**NE#9**. P. 3, L. 16-17: "dataset" should be more specific, such as "DTM with a grid size of 2 m".

**Response:** done with grid size of 2m is added.

**NE#10**. P. 3, L. 24-17: "the 5m grid DTM… by the authors." This portion seems unnecessary if the 5-m DEM by such aerial photogrammetry has never used for this study. Please clarify and remove this part if applicable.

**Response:** Please read back what was the previous reviewers query as you asked to include this paragraph in the text (see #n4.33). We do not want to remove a paragraph you asked us to add and now you asked us to remove... Please be coherent with what you asked for any revision.

**NE#11**. P. 4, L. 31: Please consider to split the section here. The former subsection (3.1) can be about the existing knowledge on geological facts, while the latter (3.2) is about the updates of the existing geological maps using the authors' methods.

**Response:** Done we added 3.1 (Hengchun Geological state of the art) and 3.2 (updated Hengchun Geology and neotectonics)

**NE#12**. P. 5, L. 23: Change "There western" to "Their western".

**Response:** "their" is added instead of there

**NE#13**. P. 6, L. 4-6: This portion should be in the subsection 3.1 (previous knowledge) as noted above.

**Response:** No we'd rather conserve the neotectonics in the paragraph that describe the activity of our new observed structures. Consequently we modified the title 3.2 in order to enhance also neotectonics.

**NE#14**. P. 6, L. 7-9: This portion seems better to be stated in the Conclusions section as a future issue.

**Response:** This paragraph is cut and place in the last sentences of the conclusion.

**NE#15**. P. 6, L. 14: Better to use double quotations for "Super Master". Also, please provide some additional explanations on this term (e.g., by just adding "as a reference").

**Response:** Done: we added "as a reference"

**NE#16**. P. 6, L. 20: "The method identifies PS pixels (more than… 20133 exactly)" can be rephrased as "The method identifies 20133 PS pixels".

**Response:** We added this sentence instead of the previous one.

**NE#17**. P. 6, L. 23: To avoid misunderstandings for readers, it would be recommended to add some notes such as "Note that this LOS data is not projected onto the vertical component." after "through time series."

**Response:** This is added

**NE#18**. P. 7, L. 24: To avoid misunderstandings, "projected on" could be rephrased with "compared with".

**Response:** Despite "compared with" is less precise than "projected on" we add it.

**NE#19**. P. 7, L. 30: ", see the colour scale" can be removed.

**Response:** It is removed.

**NE#20**. P. 8, L. 25: This section is too long can be separated into "Discussion" and "Conclusions". The lines after P. 10 L. 13 can be an independent section for "Concluding remarks".

**Response:** Following the requirement of this reviewer, we divided the discussion and conclusion paragraph and came back to the initial outline of january 31st, 2017 where on our first submitted version the discussion and the conclusion were clearly separated (see line 326 page 16 of the first submitted version!). Thanks to be coherent in your different queries.

**NE#21**. P. 10, L. 1-4: The vertical component of the tectonic activity is provided by the leveling data. However, the expression "locally confirmed by … LOS" is somewhat misunderstanding. It may be better to rephrase, such as "consistent with the LOS displacement" or else.

**Response:** We remove "locally confirmed" by "consistent with"...

**NE#22**. P. 10, L. 12: "So, in order to conclude," can be removed.

**Response:** Done. One may note it was like that in the first submitted version of January 31st, 2017 (p.17 L 327)! Please be consistent in your reviewing query…

**NE#23**. Figure 1: As noted above, the extent for Figure 1b could be expanded to cover the entire data area for InSAR as well. In the caption, replace "Fig. 1a" with "(a)", also for b and c.

**Response:** We already answer to that point we provide the PSINSAR dataset only on the studied area for common sense reasons.

**NE#24**. Figure 3: Re-order the panels as:

a b

c d

e f

g h

Also, add scales and north directions for all the panels.

**Response:** Done, see new Fig.3 we also added some active fault lines in red as required .

**NE#25**. Figure 7: Please indicate in the caption that the western margin of the geological unit (flat land) is out of the UAS data and derived from the other data sources (coarser DEMs).

**Response:** We added the following sentence: "the western extension of the Hengchun alluvial plain (8) is deduced from the 5m DEM.

**NE#26**. Figure 10: Use a citation format to represent the website and URL.

**Response:** see SSCDLA both in the figure caption and the reference list.

---

## Author Response (AR5)

We thank the Editor for the comments that allowed us to clarify and improve the contents of the manuscript.

**Comments by Editor Hayakawa:**

1: Please spell out DSM and DTM when they appear for the first time (DSM: P2 L24; DTM: P2 L15).
**Answer:** We have now defined the terms in this manuscript as suggested.

2: The paper title shows UAS DTM, but now DSM is explicitly defined as the product of UAS in the manuscript. Please change the title and any other "UAS DTM" (in the body text and figure captions) to "UAS DSM" to be consistent throughout the manuscript.
**Answer:** We have modified the title and in this manuscript as suggested. Nevertheless, part of the DTMs are used in this study.

3: Please add access date and year to the citations to webpages.
**Answer:** We have now updated the citations.

Thank you very much for your time and comments for improving the manuscript.